# Boundary phase transitions of two-dimensional quantum critical XXZ model

Hong-Hao Song and Long Zhang*

Kavli Institute for Theoretical Sciences and CAS Center for Excellence in Topological Quantum Computation, University of Chinese Academy of Sciences, Beijing 100190, China
∗ longzhang@ucas.ac.cn

January 15, 2024

### Abstract

The boundary critical behavior of the two-dimensional (2D) quantum antiferromagnetic (AF) XXZ model coupled with either a dangling spin-1/2 XXZ chain or a dangling two-leg ladder on the boundary is studied with the bosonization and renormalization group analysis. A rich boundary phase diagram is obtained in each case. In the dangling chain case, the boundary either develops a long-range AF order in the easy-plane or the easy-axis direction with extraordinary critical behavior, or has a valence bond solid order with ordinary boundary critical behavior. In the case of a dangling two-leg ladder, besides the possible easy-plane or easy-axis AF ordered phases on the boundary with extraordinary critical behavior, the boundary can form a singlet phase with ordinary critical behavior without breaking any symmetry.

## 1 Introduction

The universality of critical phenomena is one of the most exciting subjects in condensed matter and statistical physics. In a critical system with boundary, physical quantities defined on the boundary, e.g., correlation functions and magnetic susceptibilities, also show singular properties characterized by universal critical exponents, and these exponents are in general different from those in the bulk [1]. The boundary critical behavior is controlled by the renormalization group (RG) fixed points of a half-infinite critical system with boundary, and the critical exponents are determined by the scaling dimensions of boundary operators [2]. In the two-dimensional (2D) conformal field theory (CFT), the boundary operators can be constructed from the bulk via operator product expansion [3–5]. However, the boundary critical behavior is more elusive in higher dimensions.

The boundary critical behavior has attracted renewed interests recently partly inspired by the development of topological states of matter with nontrivial boundary states. The quantum critical point (QCP) of a topological phase can show unconventional boundary critical behavior [6–8], which was attributed to the interactions of the gapless boundary state with the bulk state at the QCP. Moreover, the boundary critical behavior can be adopted as a general diagnosis of "symmetry-enriched" quantum critical states [9], i.e., states that belong to the same universality class in the bulk but cannot be smoothly deformed into each other without breaking certain symmetry or crossing multicritical points [10, 11].

The universality class of boundary critical behavior can be changed by adjusting the form and strength of boundary interactions, which results into a rich boundary phase diagram. In the classical O($N$) model of the Wilson-Fisher universality, this can be achieved by tuning the interaction strength on the boundary. If the boundary interaction strength is weaker than or similar to the bulk, it inherits long-range order only below the bulk transition temperature $T_{c,b}$, and the boundary critical behavior at $T_{c,b}$ is dubbed the ordinary transition. On the other hand, strong interactions on the boundary can induce long-range order at higher temperature than $T_{c,b}$ if the boundary itself is above the lower critical dimension, then the singular

behavior of boundary physical quantities at $T_{c,b}$ is dubbed the extraordinary transition. The multicritical point separating the ordinary and the extraordinary phases is called the special transition. These boundary universality classes have been extensively studied with analytical and numerical methods for the classical O($N$) models [1, 2, 12, 13].

In 3D O($N$) model with $N > 2$, it was generally believed that the extraordinary transition is impossible, because the 2D boundary cannot form long-range order alone due to the Mermin-Wagner theorem [14]. The $N = 2$ case is more subtle. The 2D boundary with strong interactions undergoes a Berezinskii-Kosterlitz-Thouless transition above $T_{c,b}$ into a critical phase. Whether it develops truly long-range order at $T_{c,b}$ was not fully settled [12].

In 2D quantum spin models with O(3) symmetry, the quantum phase transition from a topological phase protected by the spin rotation and lattice translation symmetry to the long-range antiferromagnetic (AF) order can be realized in the spin-1/2 Heisenberg model on decorated square lattice [8] and the spin-1 coupled Haldane chains [15]. While the QCPs belong to the (2+1)D O(3) Wilson-Fisher universality class in the bulk, the boundary critical behavior in both models is distinct from the ordinary universality. Moreover, various classes of nonordinary boundary critical behavior have been found in numerical simulations of other quantum spin models with dangling spin chains or ladders [16–21]. This inspires further theoretical analysis of the boundary critical behavior of (2+1)D O($N$) universality.

In Ref. [22], the boundary critical behavior of 3D O($N$) model coupled with a 2D system on the boundary is studied with RG analysis. The boundary is described by a 2D O($N$) nonlinear $\sigma$ model. Starting from the "normal" boundary state polarized by an infinitesimal external field, it is shown that its coupling to the bulk suppresses the gapless order parameter fluctuations on the boundary, thus stabilizes the weak-coupling fixed point. Moreover, the consequent slow RG flow induces a logarithmic decay of spin correlations instead of a long-range order on the boundary, which is dubbed the extraordinary-log universality class. This peculiar behavior is observed in numerical simulations of 3D classical spin models with O(3) [23, 24] or O(2) symmetry [25, 26] and also in the 3D AF Potts model with an emergent O(2) symmetry at the critical point [27, 28].

In Ref. [29], the boundary critical behavior of a (2+1)D O(3) QCP coupled with a spin-1/2 AF Heisenberg chain is studied. The dangling spin chain is treated as a quantum critical state with the non-Abelian bosonization. Its coupling to the bulk state leads to two possible phases on the boundary: one has the valence bond solid (VBS) order spontaneously breaking the translation symmetry, while the other is argued to have long-range AF order, and there is a direct phase transition between them. The nonordinary boundary critical behavior found in numerical simulations is argued to correspond to the second phase with a weak AF order, which is confirmed in later numerical simulations [30]. Moreover, weak AF order is also observed on the boundary of a 2D quantum critical Heisenberg model with a dangling two-leg ladder [20], and a quantum XXZ model with a dangling chain [21].

In this work, we theoretically study the boundary critical behavior of the 2D quantum critical XXZ model with either a dangling spin-1/2 chain or a dangling two-leg ladder, which are illustrated in Fig. 1. In each case, we treat the dangling spin chain or the ladder with the Abelian bosonization method and take its coupling to the bulk as perturbations in the RG analysis. We find a rich boundary phase diagram in each case. The results depend on whether the system has an easy-plane or an easy-axis anisotropy, and we particularly focus on the easy-plane case, which reduces to the O(3) symmetric case if the anisotropy is turned off.

In the dangling spin chain case with easy-plane anisotropy, we find a boundary phase diagram similar to the O(3) symmetric case studied in Ref. [29]. The boundary has either a VBS order with ordinary critical behavior, or a long-range AF order with extraordinary critical behavior. In the latter case, the long-range AF order is shown based on the solution of the Callan-Symanzik equation of the spin correlation function. In the dangling ladder case, a

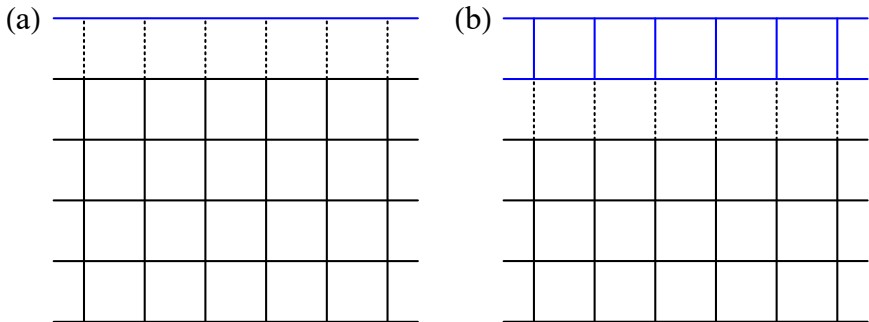

Figure 1: Schematic illustration of the 2D quantum critical XXZ model weakly coupled with (a) a dangling spin-1/2 XXZ chain, or (b) a dangling two-leg ladder. The bulk is at the quantum critical point with ordinary boundary critical behavior.

spin-singlet phase with ordinary critical behavior without breaking any symmetry emerges besides the AF order phases on the boundary. Therefore, the RG analysis is consistent with the boundary phases observed in numerical simulations of the quantum XXZ and O(3) models.

The rest of the paper is organized as follows. The boundary critical behavior of the 2D quantum critical XXZ model with a dangling spin chain and with a dangling two-leg ladder is studied in Sec. 2 and 3, respectively. In each section, we first review the bosonization and the quantum phase diagram of the spin chain or the two-leg ladder for completeness, then we consider the coupling with the bulk and the boundary phase diagram of the coupled system. We consider the effect of weak velocity difference between the bulk and the boundary modes in Sec. 4, and conclude in Sec. 5.

## 2   Boundary critical behavior with a dangling spin chain

In this section, we study the 2D quantum XXZ model with a dangling spin-1/2 chain on the boundary, which is illustrated in Fig. 1 (a). We first review the bosonization and the phase diagram of the XXZ chain for completeness, then consider its coupling to the bulk critical state and study the phase diagram of the coupled system.

### 2.1   Spin-$1/2$ XXZ chain

The Hamiltonian of the quantum XXZ chain is given by

$$H_0 = \frac{1}{2}\sum_l \left(\sigma_l^+ \sigma_{l+1}^- + \sigma_l^- \sigma_{l+1}^+\right) + \frac{\Delta}{4}\sum_l \sigma_l^z \sigma_{l+1}^z, \tag{1}$$

in which $\sigma_l^\pm = \frac{1}{2}(\sigma_l^x \pm i\sigma_l^y)$ are the spin ladder operators, and $\Delta$ is the anisotropy parameter. Applying the Jordan-Wigner transformation,

$$\sigma_l^+ = c_l^\dagger \exp\left(\pi i \sum_{j<l}(c_j^\dagger c_j + 1)\right), \tag{2}$$

$$\sigma_l^- = c_l \exp\left(-\pi i \sum_{j<l}(c_j^\dagger c_j + 1)\right), \tag{3}$$

$$\sigma_l^z = 2c_l^\dagger c_l - 1, \tag{4}$$

the spin chain is mapped to spinless fermions with nearest-neighbor interactions,

$$H_0 = -\frac{1}{2}\sum_l (c_l^\dagger c_{l+1} + \text{H.c.}) + \Delta \sum_l (c_l^\dagger c_l - 1/2)(c_{l+1}^\dagger c_{l+1} - 1/2). \tag{5}$$

Here, the case with $\Delta = 0$ corresponds to free fermions with Fermi momentum $k_\text{F} = \pi/(2a_0)$, in which $a_0$ is the lattice constant.

In the low-energy limit, we focus on fermion modes close to the Fermi points at $\pm k_\text{F}$. The right-moving modes near $+k_\text{F}$ and the left-moving modes near $-k_\text{F}$ are treated as independent fermion fields $\psi_{\text{R,L}}(x)$, thus the fermion field operator is expressed as their linear combination,

$$\psi(x) = e^{ik_\text{F}x}\psi_\text{R}(x) + e^{-ik_\text{F}x}\psi_\text{L}(x). \tag{6}$$

In the bosonization dictionary [31], the fermion field operators are expressed by

$$\psi_{\text{R,L}}(x) = \frac{1}{\sqrt{2\pi\alpha}}\exp\left(-i\sqrt{2\pi}\phi_{\text{R,L}}(x)\right), \tag{7}$$

in which the boson field operators $\phi_{\text{R,L}}(x)$ are linear combinations of boson creation and annihilation operators $b_{\text{R,L}}^\dagger$ and $b_{\text{R,L}}$,

$$\phi_{\text{R,L}}(x) = \frac{i}{\sqrt{L}}\sum_{q>0}\frac{1}{\sqrt{q}}e^{-\alpha q/2}\left(e^{\pm iqx}b_{\text{R,L}}(q) - e^{\mp iqx}b_{\text{R,L}}^\dagger(q)\right), \tag{8}$$

and $\alpha$ is a short-distance cutoff to prevent ultraviolet divergence.

Define the dual fields,

$$\phi(x) = \frac{1}{\sqrt{2}}\left(\phi_\text{L}(x) - \phi_\text{R}(x)\right), \quad \theta(x) = \frac{1}{\sqrt{2}}\left(\phi_\text{L}(x) + \phi_\text{R}(x)\right), \tag{9}$$

then they satisfy the following commutation relations,

$$[\phi(x), \partial_y\theta(y)] = i\delta(x-y), \quad [\theta(x), \partial_y\phi(y)] = i\delta(x-y). \tag{10}$$

Therefore, the conjugate momentum fields are given by $\Pi_\phi(x) = \partial_x\theta(x)$ and $\Pi_\theta(x) = \partial_x\phi(x)$, respectively.

Linearizing the free fermion dispersion relation near the Fermi points, and using the bosonization dictionary [31], the action of the spin-1/2 XXZ chain is cast into the following sine-Gordon model (see Appendix A for calculation details),

$$S_0 = \int dz d\bar{z}\left(\frac{1}{K}\partial_z\phi\partial_{\bar{z}}\phi - \lambda\cos(\sqrt{16\pi}\phi)\right), \tag{11}$$

in which $z = \tau + ix$ is the complex coordinate. Here, $K = (1 + 4\Delta/\pi)^{-1/2}$ is the Luttinger parameter of the free boson theory, and $\lambda = \Delta/(4\pi^2\alpha^2 K)$ is the interaction strength. The correlation functions at the Gaussian (free boson) fixed point are given by

$$\langle\phi(z,\bar{z})\phi(w,\bar{w})\rangle = -\frac{K}{4\pi}\ln|z-w|^2, \tag{12}$$

$$\langle\theta(z,\bar{z})\theta(w,\bar{w})\rangle = -\frac{1}{4\pi K}\ln|z-w|^2, \tag{13}$$

$$\langle\theta(z,\bar{z})\phi(w,\bar{w})\rangle = \langle\phi(z,\bar{z})\theta(w,\bar{w})\rangle = \frac{1}{4\pi}\ln\frac{z-w}{\bar{z}-\bar{w}}. \tag{14}$$

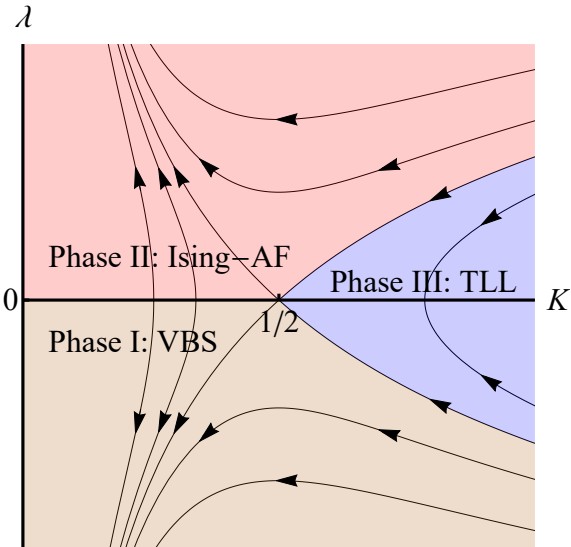

Figure 2: The RG flow diagram of the sine-Gordon model in Eq. (11). The RG flow depicts three phases shaded in different colors. The nature of each phase is analyzed in the main text.

According to the bosonization dictionary [31, 32], the spin operators are expressed by

$$\sigma^{\pm}(x) = \frac{1}{\sqrt{2\pi\alpha}} e^{\pm i\sqrt{\pi}\theta(x)}\left((-1)^{x/a_0} + e^{\pm i2\sqrt{\pi}\phi(x)}\right), \tag{15}$$

$$\sigma^{z}(x) = \frac{2}{\sqrt{\pi}}\partial_x\phi(x) + (-1)^{x/a_0}\frac{2}{\pi\alpha}\cos\left(2\sqrt{\pi}\phi(x)\right), \tag{16}$$

thus the AF order parameter is given by

$$\vec{n} = \left(\cos\left(\sqrt{\pi}\theta\right), \sin\left(\sqrt{\pi}\theta\right), \cos\left(\sqrt{4\pi}\phi\right)\right), \tag{17}$$

while the VBS order parameter can be obtained from the operator product expansion (OPE) of spin operators,

$$O_{\text{VBS}} = (-1)^l \vec{\sigma}_l \cdot \vec{\sigma}_{l+1} \simeq \sin\left(\sqrt{4\pi}\phi\right). \tag{18}$$

The phase diagram of the XXZ chain can be obtained from the RG analysis. Starting from the Gaussian fixed point, and treating the cosine term as a perturbation, the RG equations can be derived with the OPE relations given in Appendix B.1,

$$\frac{\mathrm{d}\lambda}{\mathrm{d}l} = (2 - 4K)\lambda, \tag{19}$$

$$\frac{\mathrm{d}K^{-1}}{\mathrm{d}l} = 8\pi^2\lambda^2. \tag{20}$$

Here, $l$ is the renormalization parameter and is related to the renormalization energy scale by $l \simeq \ln\mu$. The RG flow is shown in Fig. 2, and depicts three phases. In phase I and II, the interaction strength $\lambda$ flows to infinity, thus the cosine potential term in the action Eq. (11) dominates in the low-energy limit, and the $\phi$ field is pinned at the potential minima. In phase I, $\lambda \to -\infty$ in the RG flow, thus the minima $\phi = \frac{1}{2}(n + 1/2)\sqrt{\pi}$ ($n \in \mathbb{Z}$), and the chain has a long-range VBS order with $\langle O_{\text{VBS}}\rangle \neq 0$ and spontaneously breaks the translation symmetry. In phase II, $\lambda \to +\infty$ in the RG flow, and the minima $\phi = \frac{1}{2}n\sqrt{\pi}$ ($n \in \mathbb{Z}$) lead to a long-range easy-axis Ising-AF order, $\langle n_z \rangle \neq 0$. In phase III, the cosine term is irrelevant, $\lambda \to 0$, and the RG flow approaches a fixed point with a finite Luttinger parameter $K(l \to +\infty) > 1/2$. The XXZ chain forms a Tomonaga-Luttinger liquid (TLL) state described by the free boson theory.

## 2.2 Coupling to the bulk

The coupling of the spin chain to the critical fluctuations in the bulk strongly influences its properties. Denoting the AF order parameter field in the bulk by $\vec{m}$, the linear coupling between the spin chain and the bulk AF order parameters is given by

$$S_{\text{int}} = \int \mathrm{d}z \mathrm{d}\bar{z} \left( g_{xx}(n_1 m_1 + n_2 m_2) + g_z n_3 m_3 \right) \tag{21}$$

with $g_{xx} > 0$ and $g_z > 0$, which respects the O(2) spin rotation symmetry. The correlation of the bulk AF order parameter field close to the boundary has the ordinary critical behavior,

$$C_{ab}(z,\bar{z};w,\bar{w}) = \langle m_a(z,\bar{z}) m_b(w,\bar{w}) \rangle = \frac{\delta_{a,b}}{|z-w|^{3-2\epsilon_a}}, \quad a,b = 1,2,3, \tag{22}$$

in which $\epsilon_a$ is the boundary anomalous dimension of the ordinary universality class, which can be calculated perturbatively with field-theoretic methods [2]. Here, it is implicitly assumed that the velocity of the bulk critical mode is the same as the dangling spin chain. We will show that a weak velocity difference is irrelevant in the RG flow in Sec. 4.1.

The RG equations of the coupled system can be derived with the OPE relations listed in Appendix B.1,

$$\frac{\mathrm{d}\lambda}{\mathrm{d}l} = (2-4K)\lambda + \frac{1}{2}\pi g_z^2, \tag{23}$$

$$\frac{\mathrm{d}K^{-1}}{\mathrm{d}l} = 8\pi^2\lambda^2 + 2\pi^2 g_z^2 - \frac{\pi^2 g_{xx}^2}{K^2}, \tag{24}$$

$$\frac{\mathrm{d}g_{xx}}{\mathrm{d}l} = \left(\frac{1}{2} - \frac{1}{4K} + \epsilon_{xx}\right)g_{xx}, \tag{25}$$

$$\frac{\mathrm{d}g_z}{\mathrm{d}l} = \left(\frac{1}{2} - K + \epsilon_z\right)g_z + \pi\lambda g_z, \tag{26}$$

which reduces to the RG equations of the XXZ spin chain in Eqs. (19)–(20) if the couplings $g_{xx}$ and $g_z$ are turned off. Solving the coupled RG equations completely in the high-dimensional parameter space is in general quite difficult, thus we focus on the influence of weak couplings $g_{xx}$ and $g_z$ on the phases of the spin chain.

Starting from the VBS order phase of the spin chain, in which $\lambda \to -\infty$ and $K \to 0$ in the RG flow, we find both coupling strengths $g_{xx}$ and $g_z$ are renormalized to zero, thus the dangling spin chain and the critical bulk mode effectively decouple from each other. Therefore, the the VBS order and the ordinary boundary critical behavior coexist on the boundary.

Starting from the Ising-AF order phase of the spin chain, where $\lambda \to +\infty$ and $K \to 0$ in the RG flow, we find that $g_{xx}$ is renormalized to zero while $g_z$ diverges in the RG flow, and it turns out to accelerate the flow of $\lambda$ and $K$ towards infinity and zero, respectively. Therefore, the Ising-AF order on the boundary is stabilized, and induces an extraordinary boundary spin correlations in the $z$ direction. In contrast, the boundary spin correlations in the $xy$ direction retains its ordinary critical behavior, because the coupling $g_{xx}$ is renormalized to zero.

The fate of the TLL phase of the spin chain perturbed by the coupling to the bulk deserves more attention. In this phase, $\lambda$ flows to zero while $K$ flows to a finite value larger than $1/2$. According to the RG equation (26), $g_z$ is irrelevant and diminishes in the RG flow, but $g_{xx}$ is relevant. Moreover, the diverging $g_{xx}$ feeds back on the RG flow of $K$, and drives it to infinity. Therefore, the coupled RG equations accommodate a stable boundary phase with $\lambda \to 0$, $g_z \to 0$, $g_{xx} \to +\infty$ and $K \to +\infty$. A similar boundary phase was found in Ref. [29] in the O(3) symmetric case with non-Abelian bosonization, and argued to have a long-range

AF order on the boundary. We will examine the spin correlation functions with the Callan-Symanzik equation below.

The correlation functions can be calculated by integrating the Callan-Symanzik equation along the RG flow trajectory [33, 34]. For simplicity, we take the bare theory close to the Gaussian fixed point with bare coupling constants $\lambda_0 = 0$, $g_{z,0} = 0$, $K_0 = 1/2 + \epsilon_0$ with $\epsilon_0 > 0$, and $g_{xx,0} = g_0 > 0$. We denote the bare energy cutoff scale by $M$, and the renormalization energy scale by $\mu$. The coupled RG equations are simplified as

$$\frac{dK}{dl} = \mu\frac{dK}{d\mu} = \pi^2 g_{xx}^2, \tag{27}$$

$$\frac{dg_{xx}}{dl} = \mu\frac{dg_{xx}}{d\mu} = \Big(\frac{1}{2} - \frac{1}{4K} + \epsilon_{xx}\Big)g_{xx} \tag{28}$$

with the initial conditions $K(l = 0) = K_0$ and $g_{xx}(l = 0) = g_0$. The simplified RG equations have the same form as those of the sine-Gordon model and can be exactly solved. In order to focus on the asymptotic behavior of the solution in the $K \to +\infty$ regime, we drop the subleading $1/(4K)$ term in Eq. (28), and find the asymptotic solution,

$$K(l) \simeq \frac{\pi^2 g_0^2}{1 + 2\epsilon_{xx}}\big(e^{(1+2\epsilon_{xx})l} - 1\big) + K_0, \quad g_{xx}(l) = g_0 e^{(1/2+\epsilon_{xx})l}. \tag{29}$$

The two-point correlation function of a given field operator in the momentum space $G^{(2)}(p; g_{xx})$ satisfies the Callan-Symanzik equation,

$$p\frac{\partial G^{(2)}}{\partial p} - \beta(g_{xx})\frac{\partial G^{(2)}}{\partial g_{xx}} + (2 - 2\gamma)G^{(2)} = 0. \tag{30}$$

The $\beta$-function of the coupling constant $g_{xx}$ is given in Eq. (28). The anomalous dimension of the specified field operator is defined by

$$\gamma = \frac{1}{2}\frac{d\ln Z}{d\ln\mu}, \tag{31}$$

in which $Z$ is the field-strength renormalization factor defined by

$$G^{(2)}(p; g_{xx}, \mu) = Z^{-1}G^{(2)}(p; g_0, M). \tag{32}$$

From the correlation functions of the boson fields $\phi$ and $\theta$ in the free theory, Eqs. (12) and (13), we find the factors $Z_\phi = K_0/K$ and $Z_\theta = K/K_0$, thus we find $\gamma_\phi = -\pi^2 g_{xx}^2/(2K)$, and $\gamma_\theta = \pi^2 g_{xx}^2/(2K)$. The two-point correlation functions are obtained by integrating the Callan-Symanzik equation along the RG flow trajectory detailed in Appendix C,

$$G_\phi^{(2)}(p) = -\frac{1}{4\pi K_0}\frac{1}{p^2}\Big(\frac{\pi^2 g_0^2\big((\mu/p)^{1+2\epsilon_{xx}} - 1\big)}{1 + 2\epsilon_{xx}} + K_0\Big)^2, \tag{33}$$

$$G_\theta^{(2)}(p) = -\frac{K_0}{4\pi}\frac{1}{p^2}\Big(\frac{\pi^2 g_0^2\big((\mu/p)^{1+2\epsilon_{xx}} - 1\big)}{1 + 2\epsilon_{xx}} + K_0\Big)^{-2}. \tag{34}$$

In the low-energy limit $p/\mu \ll 1$, the correlation functions have the asymptotic form,

$$G_\phi^{(2)}(p) \simeq -\frac{1}{4\pi K_0}\frac{(\pi^2 g_0^2)^2}{(1 + 2\epsilon_{xx})^2}\frac{\mu^{2(1+2\epsilon_{xx})}}{p^{4(1+\epsilon_{xx})}}, \tag{35}$$

$$G_\theta^{(2)}(p) \simeq -\frac{K_0}{4\pi}\frac{(1 + 2\epsilon_{xx})^2}{(\pi^2 g_0^2)^2}\frac{p^{4\epsilon_{xx}}}{\mu^{2(1+2\epsilon_{xx})}}. \tag{36}$$

In the real space, we find

$$G_\phi^{(2)}(r) = -\frac{A}{4\pi K_0}\frac{(\pi^2 g_0^2)^2}{(1+2\epsilon_{xx})^2}(\mu r)^{2(1+2\epsilon_{xx})},$$

(37)

$$G_\theta^{(2)}(r) = -\frac{BK_0}{4\pi}\frac{(1+2\epsilon_{xx})^2}{(\pi^2 g_0^2)^2}(\mu r)^{-2(1+2\epsilon_{xx})},$$

(38)

in which

$$A = \frac{2^{-4\epsilon_{xx}-3}\pi}{\sin(2\epsilon_{xx}\pi)\Gamma(2\epsilon_{xx}+2)^2}, \quad B = -\frac{2^{4\epsilon_{xx}+1}\sin(2\epsilon_{xx}\pi)\Gamma(2\epsilon_{xx}+1)^2}{\pi}$$

(39)

are constant factors. The correlation functions of the vertex operators satisfy

$$\left\langle e^{2i\sqrt{\pi}\phi(x)}e^{-2i\sqrt{\pi}\phi(y)}\right\rangle \simeq e^{2\pi\langle\phi(x)\phi(y)\rangle} \to 0, \quad |x-y| \to \infty,$$

(40)

$$\left\langle e^{i\sqrt{\pi}\theta(x)}e^{-i\sqrt{\pi}\theta(y)}\right\rangle \simeq e^{\frac{1}{2}\pi\langle\theta(x)\theta(y)\rangle} \to 1, \quad |x-y| \to \infty.$$

(41)

Therefore, the boundary has long-range easy-plane AF order, but does not show VBS or easy-axis AF order.

The coupled RG equations (23)–(26) have a new fixed point,

$$K = \frac{1}{2}-\epsilon_{xx}, \lambda = -\frac{\epsilon_{xx}+\epsilon_z}{\pi}, g_{xx} = \frac{\sqrt{2(\epsilon_{xx}+\epsilon_z)(3\epsilon_{xx}+\epsilon_z)}}{\pi}, g_z = \frac{\sqrt{8\epsilon_{xx}(\epsilon_{xx}+\epsilon_z)}}{\pi}$$

(42)

up to the first order in $\epsilon_{xx}$ and $\epsilon_z$, which controls the transition from the VBS order to the AF order on the boundary. In particular, if $\epsilon_{xx} = \epsilon_z$, the fixed point satisfies $g_{xx} = g_z$ thus has the O(3) symmetry, and corresponds to the one found in Ref. [29]. Linearizing the RG equations near the fixed point and finding the scaling dimensions of the eigenoperators under an infinitesimal scale transformation, we find two scaling dimensions are positive. This indicates that the new fixed point has two relevant directions, thus the boundary transition from the AF order to the VBS order in the XXZ model with a dangling spin chain is in general first-order. However, in the O(3) symmetric case, only one relevant boundary operator is allowed by the O(3) symmetry, thus the boundary transition between the AF order and the VBS order is continuous, which has been proposed in Ref. [29] and observed in recent numerical simulations [30].

## 3   Boundary critical behavior with a dangling two-leg ladder

In this section, we study the boundary critical behavior of the 2D quantum XXZ model with a dangling two-leg ladder, which is illustrated in Fig. 1 (b). The same strategy as in the dangling spin chain case is adopted. We first depict the phase diagram of the two-leg ladder, then examine the stability of each phase against a weak coupling to the critical bulk states.

### 3.1   Two-leg ladder

The Hamiltonian of the spin-1/2 two-leg ladder is given by $H_0 = H_\sigma + H_\tau + H_{\sigma\tau}$. Here, the two spin chains are described by

$$H_\sigma = \frac{1}{2}\sum_l(\sigma_l^+\sigma_{l+1}^- + \sigma_l^-\sigma_{l+1}^+) + \frac{\Delta}{4}\sum_l \sigma_l^z\sigma_{l+1}^z,$$

(43)

$$H_\tau = \frac{1}{2}\sum_l(\tau_l^+\tau_{l+1}^- + \tau_l^-\tau_{l+1}^+) + \frac{\Delta}{4}\sum_l \tau_l^z\tau_{l+1}^z,$$

(44)

and $H_{\sigma\tau}$ is the interchain coupling term,

$$H_{\sigma\tau} = \frac{1}{2}\lambda_{xy}\sum_l(\sigma_l^+\tau_l^- + \sigma_l^-\tau_l^+) + \frac{1}{4}\lambda_z\sum_l\sigma_l^z\tau_l^z. \tag{45}$$

A set of boson fields $\phi_i$ and $\theta_i$ ($i = 1, 2$) is introduced for each spin chain, thus the boson Hamiltonians in the continuum limit are given by

$$H_{\sigma,\tau} = \int dx\left(\frac{1}{2K^2}(\partial_x\phi_{1,2})^2 + \frac{1}{2}(\partial_x\theta_{1,2})^2 - \frac{\Delta}{2\pi^2\alpha^2}\cos\left(\sqrt{16\pi}\phi_{1,2}\right)\right), \tag{46}$$

in which $K = (1 + 4\Delta/\pi)^{-1/2}$ is the Luttinger parameter. The interchain coupling term Eq. (45) is bosonized with the spin operators in Eqs. (15) and (16),

$$\tau^+(x)\sigma^-(x) + \text{H.c.} \simeq \frac{1}{\pi\alpha}\cos\left(\sqrt{\pi}(\theta_1 - \theta_2)\right) + \frac{1}{\pi\alpha}\cos\left(\sqrt{\pi}(\theta_1 - \theta_2 + 2\phi_1 - 2\phi_2)\right), \tag{47}$$

$$\tau^z(x)\sigma^z(x) \simeq \frac{4}{\pi}\partial\phi_1\partial\phi_2 + \frac{2}{\pi^2\alpha^2}\left(\cos\left(2\sqrt{\pi}(\phi_1 + \phi_2)\right) + \cos\left(2\sqrt{\pi}(\phi_1 - \phi_2)\right)\right). \tag{48}$$

Introducing the symmetric and antisymmetric combinations of boson fields, $\phi_{S,A} = \frac{1}{\sqrt{2}}(\phi_1\pm\phi_2)$ and $\theta_{S,A} = \frac{1}{\sqrt{2}}(\theta_1 \pm \theta_2)$, and retaining only the most relevant potential terms, we find the following bosonized Hamiltonian of the two-leg ladder in the continuum limit [35–39],

$$H_0 = \int dx\left(\sum_{i=S,A}\left(\frac{1}{2K_i^2}(\partial_x\phi_i)^2 + \frac{1}{2}(\partial_x\theta_i)^2\right) + 2g_1\cos\left(\sqrt{2\pi}\theta_A\right)\right.$$
$$\left. + 2g_2\cos\left(2\sqrt{2\pi}\phi_A\right) + 2g_3\cos\left(2\sqrt{2\pi}\phi_S\right)\right), \tag{49}$$

in which the coupling constants are defined by $K_{S,A} = \left(1 + \frac{4\Delta}{\pi} \pm \frac{\lambda_z}{2\pi}\right)^{-1/2}$, $g_1 = \pi\lambda_{xy}/(2\pi a_0)^2$, and $g_2 = g_3 = \lambda_z/(2\pi a_0)^2$. The symmetric and the antisymmetric sectors decouple from each other due to the interchain reflection symmetry. The boson fields in each sector satisfy the commutation relations,

$$[\phi_{S,A}(x), \partial_y\theta_{S,A}(y)] = i\delta(x - y), \quad [\theta_{S,A}(x), \partial_y\phi_{S,A}(y)] = i\delta(x - y). \tag{50}$$

Similar to the single spin chain case, we introduce the complex coordinates in each sector and express the action as

$$S_S = \int dz d\bar{z}\left(\frac{1}{K_S}\partial_z\phi_S\partial_{\bar{z}}\phi_S + g_3\cos\left(\sqrt{8\pi}\phi_S\right)\right), \tag{51}$$

$$S_A = \int dz d\bar{z}\left(\frac{1}{K_A}\partial_z\phi_A\partial_{\bar{z}}\phi_A + g_1\cos\left(\sqrt{2\pi}\theta_A\right) + g_2\cos\left(\sqrt{8\pi}\phi_A\right)\right). \tag{52}$$

We note that the action of the ladder is not Lorentz invariant in general, because the symmetric and the antisymmetric sectors have different velocities, thus we use different definitions of complex coordinates in the two sectors implicitly above. In Sec. 4.2, we will show that the velocity difference diminishes in the RG flow. The correlation functions at the Gaussian fixed point are given by

$$\langle\phi_{S,A}(z,\bar{z})\phi_{S,A}(w,\bar{w})\rangle = -\frac{K_{S,A}}{4\pi}\ln|z - w|^2, \tag{53}$$

$$\langle\theta_{S,A}(z,\bar{z})\theta_{S,A}(w,\bar{w})\rangle = -\frac{1}{4\pi K_{S,A}}\ln|z - w|^2, \tag{54}$$

$$\langle\theta_{S,A}(z,\bar{z})\phi_{S,A}(w,\bar{w})\rangle = \langle\phi_{S,A}(z,\bar{z})\theta_{S,A}(w,\bar{w})\rangle = \frac{1}{4\pi}\ln\frac{z - w}{\bar{z} - \bar{w}}. \tag{55}$$

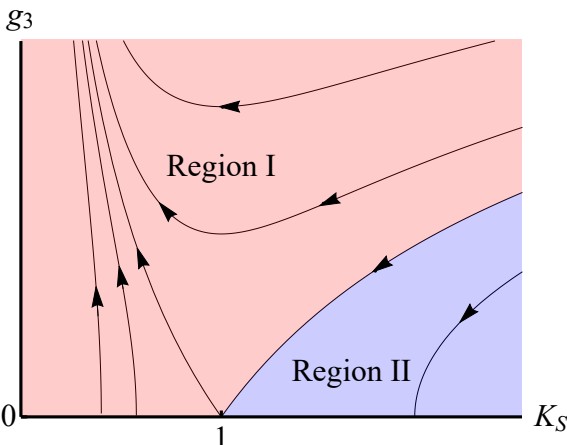

Figure 3: The RG flow of the symmetric sector action Eq. (51) in the two-leg ladder. Two different regions are shaded in different colors. The nature of each region is analyzed in the main text.

The RG equations of the ladder can be derived from the OPE relations listed in Appendix B.2. The symmetric sector is described by a sine-Gordon model, thus the RG equations are given by

$$\frac{\mathrm{d}K_\mathrm{S}}{\mathrm{d}l} = -4\pi^2 K_\mathrm{S}^2 g_3^2, \tag{56}$$

$$\frac{\mathrm{d}g_3}{\mathrm{d}l} = (2 - 2K_\mathrm{S})g_3, \tag{57}$$

and the RG flow is plotted in Fig. 3. In region I, $g_3$ is relevant and flows to infinity, while $K_\mathrm{S}$ vanishes in the RG flow, thus $\phi_\mathrm{S}$ is pinned at the minima of the cosine potential, $\phi_\mathrm{S} = \sqrt{\pi/2}(n+1/2)$ ($n \in \mathbb{Z}$), while correlation functions of vertex operators of the dual field $\theta_\mathrm{S}$ decay exponentially. In region II, $g_3$ vanishes and $K_\mathrm{S}$ flows to a finite value larger than one, thus the boson fields $\phi_\mathrm{S}$ and $\theta_\mathrm{S}$ remain critical.

In the antisymmetric sector, the RG equations are given by

$$\frac{\mathrm{d}K_\mathrm{A}}{\mathrm{d}l} = \pi^2 g_1^2 - 4\pi^2 K_\mathrm{A}^2 g_2^2, \tag{58}$$

$$\frac{\mathrm{d}g_1}{\mathrm{d}l} = \Big(2 - \frac{1}{2K_\mathrm{A}}\Big)g_1, \tag{59}$$

$$\frac{\mathrm{d}g_2}{\mathrm{d}l} = (2 - 2K_\mathrm{A})g_2. \tag{60}$$

We find at least one of the cosine terms is relevant and pins the corresponding boson field $\phi_\mathrm{A}$ or $\theta_\mathrm{A}$ at the minima of the potential term. In the first case, $g_2$ flows to infinity, thus $\phi_\mathrm{A}$ is pinned at $\sqrt{\pi/2}(n+1/2)$ ($n \in \mathbb{Z}$), while $K_\mathrm{A}$ and $g_1$ vanish in the RG flow. In the second case, $g_1$ flows to infinity, thus $\theta_\mathrm{A}$ is pinned at $\sqrt{2\pi}(n + 1/2)$ ($n \in \mathbb{Z}$), while $K_\mathrm{A}$ flows to infinity and $g_2$ vanishes.

Combining the results of the symmetric and the antisymmetric sectors, we find four phases of the two-leg ladder. In order to characterize these phases, we define $\vec{S} = \vec{\sigma} - \vec{\tau}$ and express

it in terms of boson fields,

$$S^+(x) = \frac{2i}{\sqrt{2\pi\alpha}} e^{i\sqrt{\pi/2}\theta_S(x)} \left( e^{i\sqrt{2\pi}\phi_S(x)} \sin\left(\sqrt{\pi/2}\theta_A(x) + \sqrt{2\pi}\phi_A(x)\right) \right.$$

$$\left. + (-1)^{x/a_0} \sin\left(\sqrt{\pi/2}\theta_A(x)\right) \right), \tag{61}$$

$$S^z(x) = \sqrt{\frac{8}{\pi}}\partial_x\phi_A - (-1)^{x/a_0}\frac{4}{\pi\alpha}\sin\left(\sqrt{2\pi}\phi_S\right)\sin\left(\sqrt{2\pi}\phi_A\right). \tag{62}$$

*Ising-AF phase.* (Symmetric sector: $g_3 \to +\infty$, $K_S \to 0$; Antisymmetric sector: $g_1 \to 0$, $g_2 \to +\infty$, $K_A \to 0$.) The boson fields $\phi_S$ and $\phi_A$ are pinned at the minima of their cosine potential terms, and the correlation functions of the vertex operators of the dual fields $\theta_S$ and $\theta_A$ decay exponentially. Therefore, there is a long-range easy-axis AF order, $\langle(-1)^{x/a_0}S^z(x)\rangle \neq 0$, and the inplane spin correlation function $\langle S^+S^- + S^-S^+\rangle$ decays exponentially.

*Singlet phase.* ($g_3 \to +\infty$, $K_S \to 0$; $g_1 \to +\infty$, $g_2 \to 0$, $K_A \to +\infty$.) The boson fields $\phi_S$ and $\theta_A$ are pinned at the minima of their corresponding cosine potential terms, and the correlations of the vertex operators of their dual fields $\theta_S$ and $\phi_A$ decay exponentially. Therefore, we find both spin correlation functions $\langle S^zS^z\rangle$ and $\langle S^+S^- + S^-S^+\rangle$ decay exponentially, and this phase is a disordered phase.

*XY1 phase.* ($g_3 \to 0$, $K_S \to$ finite value; $g_1 \to +\infty$, $g_2 \to 0$, $K_A \to +\infty$.) The boson field $\theta_A$ is pinned, while $\phi_S$ and $\theta_S$ remain critical. Therefore, $\langle S^zS^z\rangle$ decays exponentially, while the AF component of the inplane spin correlation function $\langle S^+S^- + S^-S^+\rangle$ decays in power law.

*XY2 phase.* ($g_3 \to 0$, $K_S \to$ finite value; $g_1 \to 0$, $g_2 \to +\infty$, $K_A \to 0$.) The boson field $\phi_A$ is pinned, while $\phi_S$ and $\theta_S$ remain critical. Therefore, $\langle S^+S^- + S^-S^+\rangle$ decays exponentially, while the AF component of $\langle S^zS^z\rangle$ and $\langle(S^+)^2(S^-)^2 + (S^-)^2(S^+)^2\rangle$ decay in power law.

## 3.2   Coupling to the bulk

In this section, we shall study the influence of the coupling to the bulk critical AF modes on the dangling two-leg ladder. We adopt the same strategy as the dangling spin chain case to examine the stability of the four phases of the two-leg ladder against a weak coupling to the bulk.

The AF order parameters on the two chains are expressed in the boson fields by

$$\vec{n}^{\sigma,\tau} = \left( \cos\left(\sqrt{\pi/2}(\theta_S \pm \theta_A)\right), \sin\left(\sqrt{\pi/2}(\theta_S \pm \theta_A)\right), \cos\left(\sqrt{2\pi}(\phi_S \pm \phi_A)\right) \right). \tag{63}$$

Denoting the bulk AF order parameter field by $\vec{m}$, the coupling of the two-leg ladder and the bulk mode is given by

$$S_{\text{int}} = \int dz d\bar{z} \sum_{i=\sigma,\tau} \left( g_{xx}^i(n_1^i m_1 + n_2^i m_2) + g_z^i n_3^i m_3 \right). \tag{64}$$

Here, the effective coupling of the lower spin chain to the bulk is antiferromagnetic with $g_{xx}^\sigma > 0$ and $g_z^\sigma > 0$, while that of the upper spin chain is ferromagnetic with $g_{xx}^\tau < 0$ and $g_z^\tau < 0$. We will neglect the velocity difference between the two spin chains and the bulk for the moment and turn to this issue in Sec. 4.2.

The RG equations can be derived from the OPE relations listed in Appendix B.2. Defining

$g_i^\pm = g_i^\sigma \pm g_i^\tau$ with $i = xx, z$, we find

$$\frac{dK_S}{dl} = -4\pi^2 K_S^2 g_3^2 - \frac{1}{2}\pi^2 K_S^2 \big((g_z^+)^2 + (g_z^-)^2\big) + \frac{1}{4}\pi^2\big((g_{xx}^+)^2 + (g_{xx}^-)^2\big), \tag{65}$$

$$\frac{dK_A}{dl} = \pi^2 g_1^2 - 4\pi^2 K_A^2 g_2^2 - \frac{1}{2}\pi^2 K_A^2\big((g_z^+)^2 + (g_z^-)^2\big) + \frac{1}{4}\pi^2\big((g_{xx}^+)^2 + (g_{xx}^-)^2\big), \tag{66}$$

$$\frac{dg_1}{dl} = \Big(2 - \frac{1}{2K_A}\Big)g_1 - \frac{1}{2}\pi\big((g_{xx}^+)^2 - (g_{xx}^-)^2\big), \tag{67}$$

$$\frac{dg_2}{dl} = (2 - 2K_A)g_2 - \frac{1}{4}\pi\big((g_z^+)^2 - (g_z^-)^2\big), \tag{68}$$

$$\frac{dg_3}{dl} = (2 - 2K_S)g_3 - \frac{1}{4}\pi\big((g_z^+)^2 - (g_z^-)^2\big), \tag{69}$$

$$\frac{dg_{xx}^+}{dl} = \Big(\frac{1}{2} - \frac{1}{8}\Big(\frac{1}{K_S} + \frac{1}{K_A}\Big) + \epsilon_{xx} - \pi g_1\Big)g_{xx}^+, \tag{70}$$

$$\frac{dg_{xx}^-}{dl} = \Big(\frac{1}{2} - \frac{1}{8}\Big(\frac{1}{K_S} + \frac{1}{K_A}\Big) + \epsilon_{xx} + \pi g_1\Big)g_{xx}^-, \tag{71}$$

$$\frac{dg_z^+}{dl} = \Big(\frac{1}{2} - \frac{1}{2}(K_S + K_A) + \epsilon_z - \pi g_2 - \pi g_3\Big)g_z^+, \tag{72}$$

$$\frac{dg_z^-}{dl} = \Big(\frac{1}{2} - \frac{1}{2}(K_S + K_A) + \epsilon_z + \pi g_2 + \pi g_3\Big)g_z^-. \tag{73}$$

Let us examine the stability of the four phases of the two-leg ladder in turn.

*Singlet phase.* In this phase, $K_S \to 0$, $K_A \to +\infty$, $g_1 \to 0$, $g_2 \to +\infty$, and $g_3 \to +\infty$ in the ladder. It is easy to see that $g_{xx}^+$ and $g_z^+$ are irrelevant and vanish in the RG flow. In order to find out the fate of other coupling constants, we take a closer look at the asymptotic form of the RG equations,

$$\frac{dK_S^{-1}}{dl} \simeq 4\pi^2 g_3^2, \quad \frac{dg_3}{dl} \simeq 2g_3, \quad \frac{dg_1}{dl} \simeq 2g_1, \tag{74}$$

thus we find $g_1 \sim e^{2l}$ while $K_S^{-1} \sim e^{4l}$ diverges even faster, and $g_{xx}^-$ vanishes in the RG flow. With similar analysis, we find $g_z^-$ vanishes as well. Therefore, all coupling terms between the ladder and the bulk are irrelevant, and the boundary retains the ordinary critical behavior without breaking any symmetry.

*Ising-AF phase.* In this phase, $K_S \to 0$, $K_A \to 0$, $g_1 \to 0$, $g_2 \to +\infty$, and $g_3 \to +\infty$ in the ladder. It is easy to see that $g_{xx}^\pm$ and $g_z^+$ vanish, while $g_z^-$ increases in the RG flow, which turns out to stabilize the original RG flow of the coupling terms in the ladder. Therefore, the coupling terms $g_{xx}^\pm$ are irrelevant and the inplane spin correlation on the boundary remains in the ordinary universality class, while it develops long-range easy-axis AF order with an extraordinary critical behavior.

*XY2 phase.* In this phase, $K_S \to$ finite value, $K_A \to 0$, $g_1 \to 0$, $g_2 \to +\infty$, and $g_3 \to 0$ in the ladder. It is easy to see that $g_{xx}^\pm$ and $g_z^+$ vanish in the RG flow, but $g_z^-$ is relevant, and it turns out to drive $g_2$ to infinity, and $K_S$ and $K_A$ to zero. Therefore, the power-law correlation of $S^z$ on the boundary is strengthened to form long-range order, and this phase is unstable upon coupling to the bulk and reduces to the Ising-AF case.

*XY1 phase.* In this phase, $K_S \to$ finite value, $K_A \to +\infty$, $g_1 \to +\infty$, $g_2 \to 0$, and $g_3 \to 0$ in the ladder. It is easy to see that $g_z^\pm$ and $g_{xx}^+$ vanish in the RG flow, but $g_{xx}^-$ is relevant, and it turns out to drive $K_S$ to infinity. This case closely resembles the easy-plane AF order phase of the dangling spin chain case, but the divergence of coupling constants is even stronger. Setting

$g_{xx}^+$, $g_z^\pm$, $g_2$ and $g_3$ to be zero, we find the following asymptotic form of the RG equations,

$$\frac{\mathrm{d}K_S}{\mathrm{d}l} \simeq \frac{1}{4}\pi^2(g_{xx}^-)^2, \tag{75}$$

$$\frac{\mathrm{d}K_A}{\mathrm{d}l} \simeq \pi^2 g_1^2 + \frac{1}{4}\pi^2(g_{xx}^-)^2, \tag{76}$$

$$\frac{\mathrm{d}g_1}{\mathrm{d}l} \simeq 2g_1 + \frac{1}{2}\pi(g_{xx}^-)^2, \tag{77}$$

$$\frac{\mathrm{d}g_{xx}^-}{\mathrm{d}l} \simeq \pi g_1 g_{xx}^-. \tag{78}$$

The coupling constants $K_{S,A}$, $g_1$ and $g_{xx}^-$ diverge at a finite renormalization scale $l_0 < +\infty$ implying the inadequacy of the perturbative RG analysis, thus the Callan-Symanzik equation cannot be integrated to find the asymptotic behavior of the correlation functions. Nonetheless, the strong divergence of $K_S$ and $K_A$ suggest that both dual fields $\theta_S$ and $\theta_A$ are pinned at static values, thus the boundary has a long-range easy-plane AF order, and shows extraordinary critical behavior.

## 4 Irrelevance of weak velocity difference

In this section, we study the effect of the velocity difference of the bulk critical AF mode and the low-energy boson modes in the dangling spin chain or the two-leg ladder. Following the strategy of Ref. [29], we will show that a weak velocity difference between the boundary spin chain and the bulk decreases in the RG transformations, and the phases obtained in previous sections are stable against the weak velocity difference.

### 4.1 The dangling spin chain case

Suppose that the velocity of the bulk AF mode $\vec{m}$ differs from that of the dangling spin chain by a factor $1 + \delta v$, and set the spin chain velocity as the unity and define the complex coordinate $z = \tau + ix$, then the correlation function of the bulk mode is given by

$$C_{ab}(x, \tau; x', \tau') = \frac{\delta_{ab}}{\left((1 - \delta v/2)^2(x - x')^2 + (1 + \delta v/2)^2(\tau - \tau')^2\right)^{3/2 - \epsilon_a}}, \tag{79}$$

which can be expressed in the complex coordinates up to the leading order of $\delta v$ as

$$C_{ab}(z, \bar{z}; w, \bar{w}) \simeq \frac{\delta_{ab}}{|z - w|^{3 - 2\epsilon_a}} - \frac{1}{4}(3 - 2\epsilon_a)\delta v \frac{(z - w)^2 + (\bar{z} - \bar{w})^2}{|z - w|^{5 - 2\epsilon_a}}. \tag{80}$$

In the RG transformation, the $\delta v$ term produces extra bilinear terms of the boson fields in the dangling spin chain (see Appendix B.3 for calculation details). When the renormalization scale changes by $\mathrm{d}l$, the transformed free boson action is given by

$$\int \mathrm{d}x\mathrm{d}\tau \left(\left(\frac{1}{K} + \frac{1}{4}(3 - 2\epsilon_z)\pi^2 g_z^2 \delta v \mathrm{d}l + \frac{1}{8K^2}(3 - 2\epsilon_{xx})\pi^2 g_{xx}^2 \delta v \mathrm{d}l\right)(\partial_x \phi)^2 \right.$$
$$\left. + \left(\frac{1}{K} - \frac{1}{4}(3 - 2\epsilon_z)\pi^2 g_z^2 \delta v \mathrm{d}l - \frac{1}{8K^2}(3 - 2\epsilon_{xx})\pi^2 g_{xx}^2 \delta v \mathrm{d}l\right)(\partial_\tau \phi)^2\right), \tag{81}$$

thus the boson field velocity in the dangling spin chain is renormalized into

$$1 + \left(\frac{\pi^2(3 - 2\epsilon_z)K g_z^2}{4} + \frac{\pi^2(3 - 2\epsilon_{xx})g_{xx}^2}{8K}\right)\delta v \mathrm{d}l. \tag{82}$$

The ratio of the bulk velocity and the spin chain velocity changes by

$$1 + \delta v \to \frac{1 + \delta v}{1 + \left( \frac{\pi^2 (3 - 2\epsilon_z) K g_z^2}{4} + \frac{\pi^2 (3 - 2\epsilon_{xx}) g_{xx}^2}{8K} \right) \delta v \mathrm{d}l}, \tag{83}$$

and we find the RG equation of $\delta v$,

$$\frac{\mathrm{d}\delta v}{\mathrm{d}l} = -\left( \frac{\pi^2 (3 - 2\epsilon_z) K g_z^2}{4} + \frac{\pi^2 (3 - 2\epsilon_{xx}) g_{xx}^2}{8K} \right) \delta v, \tag{84}$$

which shows that the velocity difference decreases in the RG flow. In particular, the velocity difference $\delta v$ is irrelevant at the new fixed point in Eq. (42) and the coupled system has an emergent Lorentz symmetry.

Away from this fixed point, the fate of the velocity difference depends on the asymptotic behavior of $K g_z^2$ and $K^{-1} g_{xx}^2$ in the RG flow. In the easy-plane AF ordered phase on the boundary, $v \to 0$, $g_z \to 0$, $g_{xx} \to +\infty$, and $K \to +\infty$, we find

$$\frac{\mathrm{d}(K^{-1} g_{xx}^2)}{\mathrm{d}l} \simeq K^{-1} g_{xx}^2 \left( 1 + 2\epsilon_{xx} - \pi^2 K^{-1} g_{xx}^2 \right), \tag{85}$$

thus $K^{-1} g_{xx}^2$ flows to a finite fixed point $(1 + 2\epsilon_{xx})/\pi^2 > 0$, while

$$\frac{\mathrm{d}(K g_z^2)}{\mathrm{d}l} \simeq (\pi^2 K^{-1} g_{xx}^2 - 2K) K g_z^2, \tag{86}$$

thus $K g_z^2$ flows to zero. Therefore, $\delta v$ is irrelevant and flows to zero in this phase, and there is an emergent Lorentz symmetry. We can similarly show that $\delta v \to 0$ in the Ising-AF ordered phase as well. In the VBS phase, we find both $K g_z^2$ and $K^{-1} g_{xx}^2$ flow to zero, thus $\delta v$ does not vanish in the RG flow in general, which is not surprising, because the dangling spin chain and the bulk effectively decouple in this phase. Nonetheless, given the decoupled nature, the boundary VBS order and the ordinary critical behavior must also be stable against the weak velocity difference.

## 4.2 The dangling two-leg ladder case

In the two-leg ladder, the symmetric and the antisymmetric sectors decouple and have different velocities in general. When the ladder couples to the bulk critical mode, the velocity of each sector approaches the bulk velocity in the RG flow, which is similar to the dangling spin chain case. Suppose that the velocity of the bulk AF mode differs from the boson field velocity in the symmetric sector of the two-leg ladder, then in terms of the complex coordinates defined in the symmetric sector, the correlation function of the bulk AF field is given by Eqs. (79) and (80), and the $\delta v$ term contributes an extra boson bilinear term in the renormalized action,

$$\int \mathrm{d}z \mathrm{d}\bar{z} \sum_{i=\sigma,\tau} \left( -\frac{\pi^2}{16} (3 - 2\epsilon_{xx}) (g_{xx}^i)^2 \delta v \mathrm{d}l (\partial_z \theta_S \partial_z \theta_S + \partial_{\bar{z}} \theta_S \partial_{\bar{z}} \theta_S) \right.$$
$$\left. -\frac{\pi^2}{8} (3 - 2\epsilon_z) (g_z^i)^2 \delta v \mathrm{d}l (\partial_z \phi_S \partial_z \phi_S + \partial_{\bar{z}} \phi_S \partial_{\bar{z}} \phi_S) \right), \tag{87}$$

thus the RG equation of $\delta v$ is given by

$$\frac{\mathrm{d}\delta v}{\mathrm{d}l} = -\sum_{i=\sigma,\tau} \left( \frac{\pi^2 (3 - 2\epsilon_z) K (g_z^i)^2}{8} + \frac{\pi^2 (3 - 2\epsilon_{xx}) (g_{xx}^i)^2}{16K} \right) \delta v. \tag{88}$$

Table 1: Summary of possible boundary phases of the 2D quantum critical AF XXZ model weakly coupled with either a dangling spin-1/2 XXZ chain or a two-leg ladder on the boundary.

| model | phase | broken symmetry | $\langle S^z S^z \rangle$ | $\langle S^+ S^- + S^- S^+ \rangle$ |
|-------|-------|-----------------|---------------------------|--------------------------------------|
| dangling chain | Ising-AF | spin flip $\mathbb{Z}_2$ | extraordinary | ordinary |
| | easy-plane AF | spin rotation O(2) | ordinary | extraordinary |
| | VBS | lattice translation | ordinary | ordinary |
| dangling ladder | Ising-AF | spin flip $\mathbb{Z}_2$ | extraordinary | ordinary |
| | easy-plane AF | spin rotation O(2) | ordinary | extraordinary |
| | singlet | none | ordinary | ordinary |

With the similar asymptotic analysis, we find that in both the Ising-AF and the easy-plane AF ordered phases, the velocity difference is irrelevant and $\delta v \to 0$ in the RG flow, thus there is an emergent Lorentz symmetry at low energy. On the other hand, the ladder effectively decouples from the bulk in the singlet phase, and $\delta v$ decreases but does not vanish in the RG flow. The ordinary boundary critical behavior in this phase is nonetheless stable against a weak velocity difference. Similar analysis can be applied to the antisymmetric sector and leads to the same conclusion.

# 5 Summary

We studied the boundary critical behavior of a 2D quantum critical AF XXZ model coupled either to a dangling spin chain or a two-leg ladder on its boundary. In each case, we start from the bosonized effective field theory of the spin chain or the ladder, and examine the stability of phases in the presence of the coupling to critical bulk AF modes with the perturbative RG analysis. We find a rich boundary phase diagram in each case. The main results are summarized in Table 1.

# 6 Acknowledgements

L.Z. is indebted to Chengxiang Ding for helpful collaborations and discussions. This work is supported by the National Natural Science Foundation of China (Grant No. 12174387), the Chinese Academy of Sciences (Nos. YSBR-057 and JZHKYPT-2021-08, and the CAS Youth Innovation Promotion Association), and the Innovative Program for Quantum Science and Technology (No. 2021ZD0302600).

# A Bosonization of spin-$1/2$ XXZ chain

In the continuum limit, the low-energy effective Hamiltonian of the spin-$1/2$ XXZ chain with a linearized fermion dispersion relation is given by

$$
\begin{aligned}
H_0 = \int \mathrm{d}x \Big( &: \psi_R^\dagger(-i\partial_x)\psi_R : + : \psi_L^\dagger(i\partial_x)\psi_L : \\
&+ \Delta(: \psi_R^\dagger\psi_R + \psi_L^\dagger\psi_L :)^2 - \Delta(: \psi_R^\dagger\psi_L + \psi_L^\dagger\psi_R :)^2 \Big),
\end{aligned}
\tag{89}
$$

in which $:A := A - \langle A \rangle$ denotes the normal ordered operator. Using the bosonized form of the fermion field operators $\psi_{\mathrm{R,L}}(x)$ in Eq. (7), we find

$$: \psi_{\mathrm{R}}^{\dagger}(x)\psi_{\mathrm{R}}(x) : \simeq -\frac{1}{\sqrt{2\pi}}\partial_x \phi_{\mathrm{R}}, \tag{90}$$

$$: \psi_{\mathrm{L}}^{\dagger}(x)\psi_{\mathrm{L}}(x) : \simeq \frac{1}{\sqrt{2\pi}}\partial_x \phi_{\mathrm{L}}, \tag{91}$$

$$: \psi_{\mathrm{R}}^{\dagger}(x)(-i\partial_x)\psi_{\mathrm{R}}(x) : \simeq \frac{1}{2}(\partial_x \phi_{\mathrm{R}})^2, \tag{92}$$

$$: \psi_{\mathrm{L}}^{\dagger}(x)(i\partial_x)\psi_{\mathrm{L}}(x) : \simeq \frac{1}{2}(\partial_x \phi_{\mathrm{L}})^2, \tag{93}$$

$$: \psi_{\mathrm{R}}^{\dagger}(x)\psi_{\mathrm{L}}(x) + \mathrm{H.c.} : \simeq \frac{1}{\pi\alpha}\cos\left(\sqrt{2\pi}(\phi_{\mathrm{R}}-\phi_{\mathrm{L}})\right), \tag{94}$$

thus the bosonized XXZ Hamiltonian is given by

$$H_0 = \int \mathrm{d}x \left( \frac{1}{2}(\partial_x \phi_{\mathrm{R}})^2 + \frac{1}{2}(\partial_x \phi_{\mathrm{L}})^2 + \frac{1}{2}\Delta(\partial_x \phi_{\mathrm{R}} - \partial_x \phi_{\mathrm{L}})^2 - \frac{\Delta}{\pi^2\alpha^2}\cos^2\left(\sqrt{2\pi}(\phi_{\mathrm{R}}-\phi_{\mathrm{L}})\right) \right). \tag{95}$$

Defining the dual fields by $\phi = \frac{1}{\sqrt{2}}(\phi_{\mathrm{L}} - \phi_{\mathrm{R}})$ and $\theta = \frac{1}{\sqrt{2}}(\phi_{\mathrm{L}} + \phi_{\mathrm{R}})$, the Hamiltonian is written as

$$H_0 = \int \mathrm{d}x \left( \frac{1}{2K^2}(\partial_x \phi)^2 + \frac{1}{2}(\partial_x \theta)^2 - \frac{\Delta}{2\pi^2\alpha^2}\cos\left(\sqrt{16\pi}\phi\right) \right), \tag{96}$$

in which $K = (1 + 4\Delta/\pi)^{-1/2}$ is the Luttinger parameter. The dual boson fields satisfy the following commutation relations,

$$[\phi(x), \partial_y \theta(y)] = i\delta(x-y), \quad [\theta(x), \partial_y \phi(y)] = i\delta(x-y), \tag{97}$$

thus their conjugate momentum fields are given by $\Pi_\phi(x) = \partial_x \theta(x)$ and $\Pi_\theta(x) = \partial_x \phi(x)$, respectively. Performing the Legendre transformation, we find the sine-Gordon model action in the Euclidean spacetime with $\tau = it$,

$$S_0 = \int \mathrm{d}x\mathrm{d}\tau \left( \frac{1}{2K^2}(\partial_x \phi)^2 + \frac{1}{2}(\partial_\tau \phi)^2 - \frac{\Delta}{2\pi^2\alpha^2}\cos\left(\sqrt{16\pi}\phi\right) \right). \tag{98}$$

Rescaling the spacetime coordinates by $(x, \tau) \to (Kx, \tau)$, the Lorentz symmetry of the boson action is more explicit,

$$S_0 = \int \mathrm{d}x\mathrm{d}\tau \left( \frac{1}{2K}\left((\partial_x \phi)^2 + (\partial_\tau \phi)^2\right) - \frac{\Delta}{2\pi^2\alpha^2 K}\cos\left(\sqrt{16\pi}\phi\right) \right). \tag{99}$$

We also introduce the complex coordinates by $z = \tau + ix$ and $\bar{z} = \tau - ix$, with $\tau = (z + \bar{z})/2$, $x = (z - \bar{z})/(2i)$, $\partial_\tau = \partial_z + \partial_{\bar{z}}$ and $\partial_{\bar{z}} = i(\partial_z - \partial_{\bar{z}})$, then the action is cast into the following form,

$$S_0 = \int \mathrm{d}z\mathrm{d}\bar{z} \left( \frac{1}{K}\partial_z \phi \partial_{\bar{z}}\phi - \lambda\cos\left(\sqrt{16\pi}\phi\right) \right), \tag{100}$$

in which $\lambda = \Delta/(4\pi^2\alpha^2 K)$.

# B   Operator product expansion relations

## B.1   The dangling spin chain case

The low-energy effective field theory of the dangling spin-1/2 XXZ chain coupled to the critical AF mode in the bulk is given by

$$S_0 + S_{\text{int}} = \int dz d\bar{z} \left( \frac{1}{K} \partial_z \phi \partial_{\bar{z}} \phi - \lambda \cos\left(\sqrt{16\pi}\phi\right) + \left(g_{xx}(n_1 m_1 + n_2 m_2) + g_z n_3 m_3\right) \right), \quad (101)$$

and the correlation function of the bulk AF field at the boundary is

$$C_{ab}(z,\bar{z};w,\bar{w}) = \langle m_a(z,\bar{z})m_b(w,\bar{w}) \rangle = \frac{\delta_{a,b}}{|z-w|^{3-2\epsilon_a}}, \quad a,b = 1,2,3, \quad (102)$$

in which $\epsilon_a$ is the anomalous dimension of the boundary AF field in the ordinary universality class. The following operator product expansion (OPE) relations are derived at the Gaussian fixed point of the spin chain action,

$$: \cos\left(\sqrt{16\pi}\phi(z,\bar{z})\right) :: \cos\left(\sqrt{16\pi}\phi(w,\bar{w})\right) :$$
$$\simeq \frac{1}{2|z-w|^{8K}} - \frac{8\pi}{|z-w|^{8K-2}} \partial_z \phi \partial_{\bar{z}} \phi - \frac{4\pi(z-w)^2}{|z-w|^{8K}} \partial_z \phi \partial_z \phi - \frac{4\pi(\bar{z}-\bar{w})^2}{|z-w|^{8K}} \partial_{\bar{z}} \phi \partial_{\bar{z}} \phi, \quad (103)$$

in which more irrelevant terms with larger scaling dimensions than the vertex operator appearing in the action Eq. (101) are neglected.

Similarly, we find

$$: (n_1 m_1 + n_2 m_2)(z,\bar{z}) :: (n_1 m_1 + n_2 m_2)(w,\bar{w}) :$$
$$\simeq \frac{1}{|z-w|^{1/(2K)+3-2\epsilon_{xx}}} - \frac{\pi}{|z-w|^{1/(2K)+1-2\epsilon_{xx}}} \partial_z \theta \partial_{\bar{z}} \theta - \frac{\pi(z-w)^2}{2|z-w|^{1/(2K)+3-2\epsilon_{xx}}} \partial_z \theta \partial_z \theta \quad (104)$$
$$- \frac{\pi(\bar{z}-\bar{w})^2}{2|z-w|^{1/(2K)+3-2\epsilon_{xx}}} \partial_{\bar{z}} \theta \partial_{\bar{z}} \theta,$$

$$: n_3 m_3(z,\bar{z}) :: n_3 m_3(w,\bar{w}) :$$
$$\simeq \frac{1}{2|z-w|^{2K+3-2\epsilon_z}} - \frac{2\pi}{|z-w|^{2K+1-2\epsilon_z}} \partial_z \phi \partial_{\bar{z}} \phi - \frac{\pi(z-w)^2}{|z-w|^{2K+3-2\epsilon_z}} \partial_z \phi \partial_z \phi \quad (105)$$
$$- \frac{\pi(\bar{z}-\bar{w})^2}{|z-w|^{2K+3-2\epsilon_z}} \partial_{\bar{z}} \phi \partial_{\bar{z}} \phi + \frac{1}{2|z-w|^{3-2K-2\epsilon_z}} \cos\left(\sqrt{16\pi}\phi(z,\bar{z})\right),$$

$$: n_3 m_3(z,\bar{z}) :: \cos\left(\sqrt{16\pi}\phi(w,\bar{w})\right) :\simeq \frac{1}{2|z-w|^{4K}} : n_3 m_3 :, \quad (106)$$

and

$$: (n_1 m_1 + n_2 m_2)(z,\bar{z}) :: \cos\left(\sqrt{16\pi}\phi(w,\bar{w})\right) :\simeq 0. \quad (107)$$

We note that the operator $\partial_z \theta \partial_{\bar{z}} \theta$ appearing in the OPE can be replaced by $K^{-2}\partial_z \phi \partial_{\bar{z}} \phi$ due to the conjugation relation of $\phi$ and $\theta$ fields.

## B.2   The dangling two-leg ladder case

The low-energy effective field theory of the two-leg ladder coupled to the critical AF mode in the bulk is given by

$$
\begin{aligned}
S_0 + S_{\text{int}} = &\int \mathrm{d}z \mathrm{d}\bar{z} \left( \frac{1}{K_S} \partial_z \phi_S \partial_{\bar{z}} \phi_S + \frac{1}{K_A} \partial_z \phi_A \partial_{\bar{z}} \phi_A \right) \\
&+ \int \mathrm{d}z \mathrm{d}\bar{z} \left( g_1 \cos\left( \sqrt{2\pi} \theta_A \right) + g_2 \cos\left( 2\sqrt{2\pi} \phi_A \right) + g_3 \cos\left( 2\sqrt{2\pi} \phi_S \right) \right) \\
&+ \int \mathrm{d}z \mathrm{d}\bar{z} \sum_{i=\sigma,\tau} \left( g_{xx}^i (n_1^i m_1 + n_2^i m_2) + g_z^i n_3^i m_3 \right).
\end{aligned}
\tag{108}
$$

The OPE relations are listed as follows.

$$
\begin{aligned}
&: \cos\left( \sqrt{2\pi} \theta_A(z,\bar{z}) \right) :: \cos\left( \sqrt{2\pi} \theta_A(w,\bar{w}) \right) : \\
&\simeq \frac{1}{2|z-w|^{1/K_A}} - \frac{\pi}{|z-w|^{1/K_A-2}} \partial_z \theta_A \partial_{\bar{z}} \theta_A + \frac{1}{2} |z-w|^{1/K_A} : \cos\left( \sqrt{8\pi} \theta_A \right) : \\
&- \frac{\pi(z-w)^2}{2|z-w|^{1/K_A}} \partial_z \theta_A \partial_z \theta_A - \frac{\pi(\bar{z}-\bar{w})^2}{2|z-w|^{1/K_A}} \partial_{\bar{z}} \theta_A \partial_{\bar{z}} \theta_A,
\end{aligned}
\tag{109}
$$

$$
\begin{aligned}
&: \cos\left( \sqrt{8\pi} \phi_A(z,\bar{z}) \right) :: \cos\left( \sqrt{8\pi} \phi_A(w,\bar{w}) \right) : \\
&\simeq \frac{1}{2|z-w|^{4K_A}} - \frac{4\pi}{|z-w|^{4K_A-2}} \partial_z \phi_A \partial_{\bar{z}} \phi_A + \frac{1}{2} |z-w|^{4K_A} : \cos\left( \sqrt{32\pi} \phi_A \right) : \\
&- \frac{2\pi(z-w)^2}{|z-w|^{4K_A}} \partial_z \phi_A \partial_z \phi_A - \frac{2\pi(\bar{z}-\bar{w})^2}{|z-w|^{4K_A}} \partial_{\bar{z}} \phi_A \partial_{\bar{z}} \phi_A,
\end{aligned}
\tag{110}
$$

$$
\begin{aligned}
&: \cos\left( \sqrt{8\pi} \phi_S(z,\bar{z}) \right) :: \cos\left( \sqrt{8\pi} \phi_S(w,\bar{w}) \right) : \\
&\simeq \frac{1}{2|z-w|^{4K_S}} - \frac{4\pi}{|z-w|^{4K_S-2}} \partial_z \phi_S \partial_{\bar{z}} \phi_S + \frac{1}{2} |z-w|^{4K_S} : \cos\left( \sqrt{32\pi} \phi_S \right) : \\
&- \frac{2\pi(z-w)^2}{|z-w|^{4K_S}} \partial_z \phi_S \partial_z \phi_S - \frac{2\pi(\bar{z}-\bar{w})^2}{|z-w|^{4K_S}} \partial_{\bar{z}} \phi_S \partial_{\bar{z}} \phi_S,
\end{aligned}
\tag{111}
$$

$$
\begin{aligned}
&: (n_1^\sigma m_1 + n_2^\sigma m_2)(z,\bar{z}) :: (n_1^\sigma m_1 + n_2^\sigma m_2)(w,\bar{w}) : \\
&\simeq \frac{1}{|z-w|^{1/(4K_S)+1/(4K_A)+3-2\epsilon_{xx}}} \\
&- \frac{\pi}{2|z-w|^{1/(4K_S)+1/(4K_A)+1-2\epsilon_{xx}}} (\partial_z \theta_A + \partial_z \theta_S)(\partial_{\bar{z}} \theta_A + \partial_{\bar{z}} \theta_S) \\
&- \frac{\pi(z-w)^2}{4|z-w|^{1/(4K_S)+1/(4K_A)+3-2\epsilon_{xx}}} (\partial_z \theta_A + \partial_z \theta_S)(\partial_z \theta_A + \partial_z \theta_S) \\
&- \frac{\pi(\bar{z}-\bar{w})^2}{4|z-w|^{1/(4K_S)+1/(4K_A)+3-2\epsilon_{xx}}} (\partial_{\bar{z}} \theta_A + \partial_{\bar{z}} \theta_S)(\partial_{\bar{z}} \theta_A + \partial_{\bar{z}} \theta_S),
\end{aligned}
\tag{112}
$$

$$
\begin{aligned}
&: (n_3^\sigma m_3)(z,\bar{z}) :: (n_3^\sigma m_3)(w,\bar{w}) : \\
&\simeq \frac{1}{2|z-w|^{K_S+K_A+3-2\epsilon_z}} - \frac{\pi}{|z-w|^{K_A+K_S+1-2\epsilon_z}} (\partial_z \phi_A + \partial_z \phi_S)(\partial_{\bar{z}} \phi_A + \partial_{\bar{z}} \phi_S) \\
&- \frac{\pi(z-w)^2}{2|z-w|^{K_A+K_S+3-2\epsilon_z}} (\partial_z \phi_A + \partial_z \phi_S)(\partial_z \phi_A + \partial_z \phi_S) \\
&- \frac{\pi(\bar{z}-\bar{w})^2}{2|z-w|^{K_A+K_S+3-2\epsilon_z}} (\partial_{\bar{z}} \phi_A + \partial_{\bar{z}} \phi_S)(\partial_{\bar{z}} \phi_A + \partial_{\bar{z}} \phi_S) \\
&+ \frac{1}{2} |z-w|^{K_A+K_S-3+2\epsilon_z} : \cos\left( \sqrt{8\pi}(\phi_S + \phi_A) \right) :,
\end{aligned}
\tag{113}
$$

$$: (n_1^\sigma m_1 + n_2^\sigma m_2)(z,\bar{z}) :: \cos(\sqrt{2\pi}\theta_A(w,\bar{w})) :\simeq \frac{1}{2|z-w|^{1/(2K_A)}} : (n_1^\tau m_1 + n_2^\tau m_2) :, \quad (114)$$

$$: n_3^\sigma m_3(z,\bar{z}) :: \cos\left(\sqrt{8\pi}\phi_A(w,\bar{w})\right) :\simeq \frac{1}{2|z-w|^{2K_A}} : n_3^\tau m_3 :, \quad (115)$$

$$: n_3^\sigma m_3(z,\bar{z}) :: \cos(\sqrt{8\pi}\phi_S(w,\bar{w})) :\simeq \frac{1}{2|z-w|^{2K_S}} : n_3^\tau m_3 :, \quad (116)$$

$$: (n_1^\tau m_1 + n_2^\tau m_2)(z,\bar{z}) :: \cos(\sqrt{2\pi}\theta_A(w,\bar{w})) :\simeq \frac{1}{2|z-w|^{1/(2K_A)}} : n_1^\sigma m_1 + n_2^\sigma m_2 :, \quad (117)$$

$$: n_3^\tau m_3(z,\bar{z}) :: \cos(\sqrt{8\pi}\phi_A(w,\bar{w})) :\simeq \frac{1}{2|z-w|^{2K_A}} : n_3^\sigma m_3 :, \quad (118)$$

$$: n_3^\tau m_3(z,\bar{z}) :: \cos(\sqrt{8\pi}\phi_S(w,\bar{w})) :\simeq \frac{1}{2|z-w|^{2K_S}} : n_3^\sigma m_3 :, \quad (119)$$

$$: (n_1^\tau m_1 + n_2^\tau m_2)(z,\bar{z}) :: (n_1^\tau m_1 + n_2^\tau m_2)(w,\bar{w}) :$$
$$\simeq \frac{1}{|z-w|^{1/(4K_S)+1/(4K_A)+3-2\epsilon_{xx}}}$$
$$- \frac{\pi}{2|z-w|^{1/(4K_S)+1/(4K_A)+1-2\epsilon_{xx}}}(\partial_z\theta_A - \partial_z\theta_S)(\partial_{\bar{z}}\theta_A - \partial_{\bar{z}}\theta_S)$$
$$- \frac{\pi(z-w)^2}{4|z-w|^{1/(4K_S)+1/(4K_A)+3-2\epsilon_{xx}}}(\partial_z\theta_A - \partial_z\theta_S)(\partial_z\theta_A - \partial_z\theta_S)$$
$$- \frac{\pi(\bar{z}-\bar{w})^2}{4|z-w|^{1/(4K_S)+1/(4K_A)+3-2\epsilon_{xx}}}(\partial_{\bar{z}}\theta_A - \partial_{\bar{z}}\theta_S)(\partial_{\bar{z}}\theta_A - \partial_{\bar{z}}\theta_S), \quad (120)$$

$$: n_3^\tau m_3(z,\bar{z}) :: n_3^\tau m_3(w,\bar{w}) :$$
$$\simeq \frac{1}{2|z-w|^{(K_S+K_A)+3-2\epsilon_z}} - \frac{\pi}{|z-w|^{(K_S+K_A)+1-2\epsilon_z}}(\partial_z\phi_A - \partial_z\phi_S)(\partial_{\bar{z}}\phi_A - \partial_{\bar{z}}\phi_S)$$
$$- \frac{\pi(z-w)^2}{2|z-w|^{(K_S+K_A)+3-2\epsilon_z}}(\partial_z\phi_A - \partial_z\phi_S)(\partial_z\phi_A - \partial_z\phi_S)$$
$$- \frac{\pi(\bar{z}-\bar{w})^2}{2|z-w|^{(K_S+K_A)+3-2\epsilon_z}}(\partial_{\bar{z}}\phi_A - \partial_{\bar{z}}\phi_S)(\partial_{\bar{z}}\phi_A - \partial_{\bar{z}}\phi_S)$$
$$+ \frac{1}{2}|z-w|^{(K_S+K_A)-3+2\epsilon_z} : \cos(\sqrt{8\pi}(\phi_S - \phi_A)) :, \quad (121)$$

$$: (n_1^\sigma m_1 + n_2^\sigma m_2)(z,\bar{z}) :: (n_1^\tau m_1 + n_2^\tau m_2)(w,\bar{w}) :\simeq \frac{: \cos(\sqrt{2\pi}\theta_A) :}{|z-w|^{1/(4K_S)-1/(4K_A)+3-2\epsilon_{xx}}}, \quad (122)$$

and

$$: (n_3^\sigma m_3)(z,\bar{z}) :: (n_3^\tau m_3)(w,\bar{w}) :$$
$$\simeq \frac{1}{2|z-w|^{(K_A-K_S)+3-2\epsilon_z}} : \cos(\sqrt{8\pi}\phi_A) : + \frac{1}{2|z-w|^{(K_S-K_A)+3-2\epsilon_z}} : \cos(\sqrt{8\pi}\phi_S) : . \quad (123)$$

We note that crossing terms of the symmetric and the antisymmetric sectors appear in the OPE, because the two sectors are effectively coupled via the bulk. We must include these crossing terms in the effective action,

$$\gamma_\phi(\partial_z\phi_S\partial_{\bar{z}}\phi_A + \partial_{\bar{z}}\phi_S\partial_z\phi_A) + \gamma_\theta(\partial_z\theta_S\partial_{\bar{z}}\theta_A + \partial_{\bar{z}}\theta_S\partial_z\theta_A), \quad (124)$$

which produce new OPE relations,

$$: (\partial_z \phi_S \partial_{\bar{z}} \phi_A + \partial_z \phi_A \partial_{\bar{z}} \phi_S)(z, \bar{z} ) :: e^{i(\alpha \phi_S + \beta \phi_A)}(w, \bar{w}) :\simeq -\frac{\alpha \beta K_S K_A}{8\pi^2 |z - w|^2} : e^{i(\alpha \phi_S + \beta \phi_A)} :, \quad (125)$$

$$: (\partial_z \theta_S \partial_{\bar{z}} \theta_A + \partial_z \theta_A \partial_{\bar{z}} \theta_S)(z, \bar{z} ) :: e^{i(\alpha \theta_S + \beta \theta_A)}(w, \bar{w}) :\simeq -\frac{\alpha \beta}{8\pi^2 K_S K_A |z - w|^2} : e^{i(\alpha \theta_S + \beta \theta_A)} :, \quad (126)$$

$$: (\partial_z \phi_S \partial_{\bar{z}} \phi_A + \partial_z \phi_A \partial_{\bar{z}} \phi_S)(z, \bar{z} ) :: e^{i(\alpha \theta_S + \beta \theta_A)}(w, \bar{w}) :\simeq \frac{\alpha \beta}{8\pi^2 |z - w|^2} : e^{i(\alpha \theta_S + \beta \theta_A)} :, \quad (127)$$

$$: (\partial_z \theta_S \partial_{\bar{z}} \theta_A + \partial_z \theta_A \partial_{\bar{z}} \theta_S)(z, \bar{z} ) :: e^{i(\alpha \phi_S + \beta \phi_A)}(w, \bar{w}) :\simeq \frac{\alpha \beta}{8\pi^2 |z - w|^2} : e^{i(\alpha \phi_S + \beta \phi_A)} :, \quad (128)$$

and the complete RG equations are given by

$$\frac{dK_S}{dl} = -4\pi^2 K_S^2 g_3^2 - \frac{1}{2}\pi^2 K_S^2 \big((g_z^+)^2 + (g_z^-)^2\big) + \frac{1}{4}\pi^2 \big((g_{xx}^+)^2 + (g_{xx}^-)^2\big), \quad (129)$$

$$\frac{dK_A}{dl} = \pi^2 g_1^2 - 4\pi^2 K_A^2 g_2^2 - \frac{1}{2}\pi^2 K_A^2 \big((g_z^+)^2 + (g_z^-)^2\big) + \frac{1}{4}\pi^2 \big((g_{xx}^+)^2 + (g_{xx}^-)^2\big), \quad (130)$$

$$\frac{dg_1}{dl} = \Big(2 - \frac{1}{2K_A}\Big)g_1 - \frac{1}{2}\pi\big((g_{xx}^+)^2 - (g_{xx}^-)^2\big), \quad (131)$$

$$\frac{dg_2}{dl} = (2 - 2K_A)g_2 - \frac{1}{4}\pi\big((g_z^+)^2 - (g_z^-)^2\big), \quad (132)$$

$$\frac{dg_3}{dl} = (2 - 2K_S)g_3 - \frac{1}{4}\pi\big((g_z^+)^2 - (g_z^-)^2\big), \quad (133)$$

$$\frac{dg_{xx}^+}{dl} = \Big(\frac{1}{2} - \frac{1}{8}\Big(\frac{1}{K_S} + \frac{1}{K_A}\Big) + \epsilon_{xx} - \pi g_1\Big)g_{xx}^+ + \Big(\frac{1}{16 K_S K_A}\gamma_\theta - \frac{1}{16}\gamma_\phi\Big)g_{xx}^-, \quad (134)$$

$$\frac{dg_{xx}^-}{dl} = \Big(\frac{1}{2} - \frac{1}{8}\Big(\frac{1}{K_S} + \frac{1}{K_A}\Big) + \epsilon_{xx} + \pi g_1\Big)g_{xx}^- + \Big(\frac{1}{16 K_S K_A}\gamma_\theta - \frac{1}{16}\gamma_\phi\Big)g_{xx}^+, \quad (135)$$

$$\frac{dg_z^+}{dl} = \Big(\frac{1}{2} - \frac{1}{2}(K_S + K_A) + \epsilon_z - \pi g_2 - \pi g_3\Big)g_z^+ + \Big(\frac{K_S K_A}{4}\gamma_\phi - \frac{1}{4}\gamma_\theta\Big)g_z^-, \quad (136)$$

$$\frac{dg_z^-}{dl} = \Big(\frac{1}{2} - \frac{1}{2}(K_S + K_A) + \epsilon_z + \pi g_2 + \pi g_3\Big)g_z^- + \Big(\frac{K_S K_A}{4}\gamma_\phi - \frac{1}{4}\gamma_\theta\Big)g_z^+, \quad (137)$$

$$\frac{d\gamma_\phi}{dl} = \pi^2 g_z^+ g_z^-, \quad (138)$$

$$\frac{d\gamma_\theta}{dl} = \frac{\pi^2}{2} g_{xx}^+ g_{xx}^-. \quad (139)$$

They reduce to Eqs. (65)–(73) and lead to the three phases discussed in Sec. 3 if the $\gamma_\phi$ and $\gamma_\theta$ terms are neglected. In these phases, the extra terms in the RG equations produced by $\gamma_\phi$ and $\gamma_\theta$ turn out to be subleading and do not change the asymptotic RG flow of the other coupling constants. Therefore, these boundary phases are stable against the mixing of boson modes in the two sectors.

## B.3   The dangling chain case with velocity difference

In the dangling spin chain case, the velocity difference of the chain and the bulk modes leads to an extra term in the correlation function of the bulk AF fields,

$$C_{ab}(z, \bar{z}; w, \bar{w}) \simeq \frac{\delta_{ab}}{|z - w|^{3 - 2\epsilon_a}} - \frac{1}{4}(3 - 2\epsilon_a)\delta v \frac{(z - w)^2 + (\bar{z} - \bar{w})^2}{|z - w|^{5 - 2\epsilon_a}}. \quad (140)$$

This extra term modifies the following OPE relations,

$$: n_3 m_3(z,\bar{z}) :: n_3 m_3(w,\bar{w}) :$$

$$\simeq \frac{1}{2|z-w|^{2K+3-2\epsilon_z}} + \frac{1}{2|z-w|^{3-2K-2\epsilon z}} : \cos(4\sqrt{\pi}\phi) : \tag{141}$$

$$-\frac{2\pi}{|z-w|^{2K+1-2\epsilon_z}}\partial_z\phi\partial_{\bar{z}}\phi + \frac{(3-2\epsilon_z)\pi\delta v}{4|z-w|^{2K+1-2\epsilon_z}}(\partial_z\phi\partial_z\phi + \partial_{\bar{z}}\phi\partial_{\bar{z}}\phi),$$

$$: (n_1 m_1 + n_2 m_2)(z,\bar{z}) :: (n_1 m_1 + n_2 m_2)(w,\bar{w}) :$$

$$\simeq \frac{1}{|z-w|^{\frac{1}{2K}+3-2\epsilon_{xx}}} - \frac{\pi}{|z-w|^{\frac{1}{2K}+1-2\epsilon_{xx}}}\partial_z\theta\partial_{\bar{z}}\theta + \frac{(3-2\epsilon_{xx})\pi\delta v}{8|z-w|^{\frac{1}{2K}+1-2\epsilon_{xx}}}(\partial_z\theta\partial_z\theta + \partial_{\bar{z}}\theta\partial_{\bar{z}}\theta), \tag{142}$$

in which the terms proportional to $\delta v$ are not invariant under the original Lorentz transformation. Instead, under the RG transformation, these extra terms contribute boson field bilinear terms to the effective action of the dangling spin chain,

$$\int \mathrm{d}z\mathrm{d}\bar{z}\left(\frac{1}{K}\partial_z\phi\partial_{\bar{z}}\phi - \frac{1}{4}\pi^2(3-2\epsilon_z)g_z^2\delta v\mathrm{d}l(\partial_{\bar{z}}\phi\partial_{\bar{z}}\phi + \partial_z\phi\partial_z\phi)\right)$$

$$= \int \mathrm{d}x\mathrm{d}\tau\frac{1}{K}\left(\left(1+\frac{1}{4}\pi^2(3-2\epsilon_z)Kg_z^2\delta v\mathrm{d}l\right)(\partial_x\phi)^2 + \left(1-\frac{1}{4}\pi^2(3-2\epsilon_z)Kg_z^2\delta v\mathrm{d}l\right)(\partial_\tau\phi)^2\right). \tag{143}$$

Similarly, the extra terms in Eq. (142) produce the following boson bilinear terms,

$$\int \mathrm{d}z\mathrm{d}\bar{z}\left(K\partial_z\theta\partial_{\bar{z}}\theta - \frac{1}{8}\pi^2(3-2\epsilon_{xx})g_{xx}^2\delta v\mathrm{d}l(\partial_{\bar{z}}\theta\partial_{\bar{z}}\theta + \partial_z\theta\partial_z\theta)\right)$$

$$= \int \mathrm{d}x\mathrm{d}\tau K\left(\left(1+\frac{\pi^2(3-2\epsilon_{xx})g_{xx}^2\delta v\mathrm{d}l}{8K}\right)(\partial_x\theta)^2 + \left(1-\frac{\pi^2(3-2\epsilon_{xx})g_{xx}^2\delta v\mathrm{d}l}{8K}\right)(\partial_\tau\theta)^2\right). \tag{144}$$

Therefore, the velocity of the dangling spin chain is renormalized into

$$1 + \left(\frac{\pi^2(3-2\epsilon_z)Kg_z^2}{4} + \frac{\pi^2(3-2\epsilon_{xx})g_{xx}^2}{8K}\right)\delta v\mathrm{d}l. \tag{145}$$

## B.4   The dangling two-leg ladder case with velocity difference

Suppose that the velocity of the bulk AF field differs from that of the boson field in the symmetric sector of the two-leg ladder by a factor $1+\delta v$, then the correlation function of the bulk is modified as Eq. (140), and the OPE relations are modified accordingly,

$$: (n_1^\sigma m_1 + n_2^\sigma m_2)(z,\bar{z}) :: (n_1^\sigma m_1 + n_2^\sigma m_2)(w,\bar{w}) :$$

$$\simeq \frac{1}{|z-w|^{\frac{1}{4}(\frac{1}{K_S}+\frac{1}{K_A})+3-2\epsilon_{xx}}} - \frac{\pi}{2|z-w|^{\frac{1}{4}(\frac{1}{K_S}+\frac{1}{K_A})+1-2\epsilon_{xx}}}(\partial_z\theta_A + \partial_z\theta_S)(\partial_{\bar{z}}\theta_A + \partial_{\bar{z}}\theta_S)$$

$$+\frac{(3-2\epsilon_{xx})\pi\delta v}{16|z-w|^{\frac{1}{4}(\frac{1}{K_S}+\frac{1}{K_A})+1-2\epsilon_{xx}}}\left((\partial_z\theta_A + \partial_z\theta_S)(\partial_z\theta_A + \partial_z\theta_S) + (\partial_{\bar{z}}\theta_A + \partial_{\bar{z}}\theta_S)(\partial_{\bar{z}}\theta_A + \partial_{\bar{z}}\theta_S)\right), \tag{146}$$

$$: (n_3^\sigma m_3)(z,\bar{z}) :: (n_3^\sigma m_3)(w,\bar{w}) :$$

$$\simeq \frac{1}{2|z-w|^{K_S+K_A+3-2\epsilon_z}} - \frac{\pi}{|z-w|^{K_S+K_A+1-2\epsilon_z}}(\partial_z\phi_A + \partial_z\phi_S)(\partial_{\bar{z}}\phi_A + \partial_{\bar{z}}\phi_S)$$

$$+\frac{(3-2\epsilon_z)\pi\delta v}{8|z-w|^{K_S+K_A+1-2\epsilon_z}}\left((\partial_z\phi_A + \partial_z\phi_S)(\partial_z\phi_A + \partial_z\phi_S) + (\partial_{\bar{z}}\phi_A + \partial_{\bar{z}}\phi_S)(\partial_{\bar{z}}\phi_A + \partial_{\bar{z}}\phi_S)\right) \tag{147}$$

$$+\frac{1}{2}|z-w|^{K_S+K_A+3-2\epsilon_z} : \cos\left(\sqrt{8\pi}(\phi_S + \phi_A)\right) :,$$

$$: (n_1^\tau m_1 + n_2^\tau m_2)(z, \bar{z}) :: (n_1^\tau m_1 + n_2^\tau m_2)(w, \bar{w}):$$

$$\simeq \frac{1}{|z-w|^{\frac{1}{4}(\frac{1}{K_S}+\frac{1}{K_A})+3-2\epsilon_{xx}}} - \frac{\pi}{2|z-w|^{\frac{1}{4}(\frac{1}{K_S}+\frac{1}{K_A})+1-2\epsilon_{xx}}}(\partial_z\theta_A - \partial_z\theta_S)(\partial_{\bar{z}}\theta_A - \partial_{\bar{z}}\theta_S) \tag{148}$$

$$+ \frac{(3-2\epsilon_{xx})\pi\delta v}{16|z-w|^{\frac{1}{4}(\frac{1}{K_S}+\frac{1}{K_A})+1-2\epsilon_{xx}}}\Big((\partial_z\theta_A - \partial_z\theta_S)(\partial_z\theta_A - \partial_z\theta_S) + (\partial_{\bar{z}}\theta_A - \partial_{\bar{z}}\theta_S)(\partial_{\bar{z}}\theta_A - \partial_{\bar{z}}\theta_S)\Big),$$

$$: (n_3^\tau m_3)(z, \bar{z}) :: (n_3^\tau m_3)(w, \bar{w}):$$

$$\simeq \frac{1}{2|z-w|^{K_S+K_A+3-2\epsilon_z}} - \frac{\pi}{|z-w|^{K_S+K_A+1-2\epsilon_z}}(\partial_z\phi_A - \partial_z\phi_S)(\partial_{\bar{z}}\phi_A - \partial_{\bar{z}}\phi_S)$$

$$+ \frac{(3-2\epsilon_z)\pi\delta v}{8|z-w|^{K_S+K_A+1-2\epsilon_z}}\Big((\partial_z\phi_A - \partial_z\phi_S)(\partial_z\phi_A - \partial_z\phi_S) + (\partial_{\bar{z}}\phi_A - \partial_{\bar{z}}\phi_S)(\partial_{\bar{z}}\phi_A - \partial_{\bar{z}}\phi_S)\Big) \tag{149}$$

$$+ \frac{1}{2}|z-w|^{K_S+K_A+3-2\epsilon_z} : \cos\left(\sqrt{8\pi}(\phi_S-\phi_A)\right)::.$$

The extra terms from the velocity difference contribute a Lorentz symmetry violating term to the boson action in the symmetric sector,

$$-\frac{1}{16}\pi^2\delta v \mathrm{d}l \int \mathrm{d}z \mathrm{d}\bar{z}$$

$$\Big((3-2\epsilon_{xx})(g_{xx}^\sigma)^2\big(\partial_z\theta_S\partial_z\theta_S + \partial_z\theta_A\partial_z\theta_S + \partial_z\theta_S\partial_z\theta_A + \partial_{\bar{z}}\theta_S\partial_{\bar{z}}\theta_S + \partial_{\bar{z}}\theta_A\partial_{\bar{z}}\theta_S + \partial_{\bar{z}}\theta_S\partial_{\bar{z}}\theta_A\big)$$

$$+ 2(3-2\epsilon_z)(g_z^\sigma)^2\big(\partial_z\phi_S\partial_z\phi_S + \partial_z\phi_A\partial_z\phi_S + \partial_z\phi_S\partial_z\phi_A + \partial_{\bar{z}}\phi_S\partial_{\bar{z}}\phi_S + \partial_{\bar{z}}\phi_A\partial_{\bar{z}}\phi_S + \partial_{\bar{z}}\phi_S\partial_{\bar{z}}\phi_A\big)$$

$$+ (3-2\epsilon_{xx})(g_{xx}^\tau)^2\big(\partial_z\theta_S\partial_z\theta_S - \partial_z\theta_A\partial_z\theta_S - \partial_z\theta_S\partial_z\theta_A + \partial_{\bar{z}}\theta_S\partial_{\bar{z}}\theta_S - \partial_{\bar{z}}\theta_A\partial_{\bar{z}}\theta_S - \partial_{\bar{z}}\theta_S\partial_{\bar{z}}\theta_A\big)$$

$$+ 2(3-2\epsilon_z)(g_z^\tau)^2\big(\partial_z\phi_S\partial_z\phi_S - \partial_z\phi_A\partial_z\phi_S - \partial_z\phi_S\partial_z\phi_A + \partial_{\bar{z}}\phi_S\partial_{\bar{z}}\phi_S - \partial_{\bar{z}}\phi_A\partial_{\bar{z}}\phi_S - \partial_{\bar{z}}\phi_S\partial_{\bar{z}}\phi_A\big)\Big). \tag{150}$$

The RG equations imply that $g_i^+$ always flow to zero, thus $(g_i^\sigma)^2 \simeq (g_i^\tau)^2$ and the above equation can be simplified to

$$\int \mathrm{d}z \mathrm{d}\bar{z}\Big(-\frac{1}{16}\pi^2(3-2\epsilon_{xx})\big((g_{xx}^\sigma)^2+(g_{xx}^\tau)^2\big)\delta v \mathrm{d}l(\partial_z\theta_S\partial_z\theta_S + \partial_{\bar{z}}\theta_S\partial_{\bar{z}}\theta_S)$$

$$-\frac{1}{8}\pi^2(3-2\epsilon_z)\big((g_z^\sigma)^2+(g_z^\tau)^2\big)\delta v \mathrm{d}l(\partial_z\phi_S\partial_z\phi_S + \partial_{\bar{z}}\phi_S\partial_{\bar{z}}\phi_S)\Big), \tag{151}$$

thus the boson velocity in the symmetric sector is renormalized to be

$$1 + \sum_{i=\sigma,\tau}\left(\frac{\pi^2(3-2\epsilon_z)K(g_z^i)^2}{8} + \frac{\pi^2(3-2\epsilon_{xx})(g_{xx}^i)^2}{16K}\right)\delta v \mathrm{d}l. \tag{152}$$

## C   Solving the Callan-Symmanzik equation

In the momentum space, the Callan-Symanzik equation of the two-point correlation function is given by

$$p\frac{\partial G^{(2)}}{\partial p} - \beta(g_{xx})\frac{\partial G^{(2)}}{\partial g_{xx}} + (2-2\gamma)G^{(2)} = 0, \tag{153}$$

in which the $\beta$-function and the anomalous dimension $\gamma$ are defined by

$$\beta(g_{xx}) = \mu\frac{\mathrm{d}g_{xx}}{\mathrm{d}\mu}, \quad \gamma = \frac{1}{2}\frac{\mathrm{d}\ln Z}{\mathrm{d}\ln\mu}, \tag{154}$$

and $Z$ is the field-strength renormalization factor defined by

$$G^{(2)}(p; g_{xx}, \mu) = Z^{-1} G^{(2)}(p; g_0, M). \tag{155}$$

From the correlation functions of the boson fields $\phi$ and $\theta$ in the free theory, Eqs. (12) and (13), we find the factors $Z_\phi = K_0/K$ and $Z_\theta = K/K_0$, thus we find $\gamma_\phi = -\pi^2 g_{xx}^2/(2K)$, and $\gamma_\theta = \pi^2 g_{xx}^2/(2K)$.

The solution of Eq. (153) has the following form,

$$G^{(2)}(p) = \frac{1}{p^2} D(g_{xx}, K) e^{-2 \int_0^{\ln(\mu/p)} dl\, \gamma(g_{xx}, K)}, \tag{156}$$

in which the integration factors are given by

$$e^{-2 \int_0^{\ln(\mu/p)} dl\, \gamma_\phi} = \frac{\pi^2 g_{xx,0}^2 \big((\mu/p)^{1+2\epsilon_{xx}} - 1\big)}{K_0(1 + 2\epsilon_{xx})} + 1, \tag{157}$$

$$e^{-2 \int_0^{\ln(\mu/p)} dl\, \gamma_\theta} = \left( \frac{\pi^2 g_{xx,0}^2 \big((\mu/p)^{1+2\epsilon_{xx}} - 1\big)}{K_0(1 + 2\epsilon_{xx})} + 1 \right)^{-1}. \tag{158}$$

Comparing the form (156) with the correlation functions of the free boson theory,

$$G_{\phi,0}(p) = -\frac{K_0}{4\pi} \frac{1}{p^2}, \quad G_{\theta,0}(p) = -\frac{1}{4\pi K_0} \frac{1}{p^2}, \tag{159}$$

we find the factors

$$D_\phi(g_{xx}, K) = -\frac{K}{4\pi}, \quad D_\theta(g_{xx}, K) = -\frac{1}{4\pi K}. \tag{160}$$

Therefore, the renormalized two-point correlation functions are given by

$$G_\phi^{(2)}(p) = -\frac{1}{4\pi K_0} \frac{1}{p^2} \left( \frac{\pi^2 g_0^2 \big((\mu/p)^{1+2\epsilon_{xx}} - 1\big)}{1 + 2\epsilon_{xx}} + K_0 \right)^2, \tag{161}$$

$$G_\theta^{(2)}(p) = -\frac{K_0}{4\pi} \frac{1}{p^2} \left( \frac{\pi^2 g_0^2 \big((\mu/p)^{1+2\epsilon_{xx}} - 1\big)}{1 + 2\epsilon_{xx}} + K_0 \right)^{-2}. \tag{162}$$

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
