# Peer review of "Boundary phase transitions of two-dimensional quantum critical XXZ model"

_SciPost Physics_

## Round 1 · Referee Report · Anonymous (Referee 2) · 2024-3-22

Strengths

- provides analytical perspective on existing and to-be-published numerical results
- thorough calculations
- clear presentation

Weaknesses

- lack of conclusions
- lack of motivation for studying this particular model and not another one

Report

This work studies the boundary criticality of an XXZ Heisenberg model using bosonization techniques. Both the case of a dangling edge chain and a dangling edge ladder are considered, and the different RG fixed points of the boundary field theory are discussed. It is further shown that weak differences in the velocity between bulk and edge modes are not a relevant perturbation to the observed behavior.

While the calculations seem very thorough and the bosonization is a nice tool to gain analytical understanding for these systems, there are some shortcomings:

1. This work is missing a conclusion! Rather than a simple summary, the conclusion should argue what the obtained results mean in the broader context of the field. What open questions do the new results answer? Does the work open new questions for the future?

Here is just a small collection of such open questions as an inspiration:
a. What is the relation to the boundary phases found in Ref. [21]?
b. Is there something all the extraordinary/ordinary cases in Table 1 have in common? Given an arbitrary surface, can we predict (or understand) the result using some simple surface property/scaling dimension without going through the full calculation each time?
c. To my knowledge, the VBS boundary phase has remained elusive so far in numerical simulations. Can this work give us a hint why?

2. The introduction goes into a lot of detail about the previous results, however it does not answer the important question why the authors decided to study the anisotropic model. What is different in the anisotropic case and why is it exciting? What differentiates this work from Ref. [29], which did a similar calculation for the isotropic model and Ref. [21], which studied a similar model using numerics.

3. It would be helpful to cite values for $\epsilon_{xx}$ and $\epsilon_z$ or at least discuss which one is bigger in what case (easy-plane or axis) and how the RG flows depend on the values.

4. The dangling ladder can also host a Haldane phase for ferromagnetic rung coupling. Does it lead to extraordinary correlations like in the numerical simulations with a dangling $S=1$ chain?

Requested changes

1. Add a conclusion.
2. Make the introduction motivate the current work.
3. Comment more on the value of $\epsilon_{xx}$ and $\epsilon_{z}$.
4. If within the scope of the existing calculations, discuss the Haldane phase of the dangling ladder.

---

## Round 1 · Referee Report · Anonymous (Referee 1) · 2024-3-29

Strengths

detailed explanation of field theory starting points, and of most computations

detailed computations and discussions of important RG aspects (including in appendices)

Weaknesses

very unclear motivation

challenging interpretability of RG equations in the ladder case

Report

In their manuscript, Song and Zhang report on a combined bosonization and renormalization group study of boundary critical behavior in the two dimensional spin-1/2 XXZ model. Two kinds of boundaries are treated: one with a single chain, and one with an (anisotropic) spin ladder. The treatment predicts several possible scenarios, summarized in table I and whose starting RG points are also summarized in the text.

From the validity viewpoint, all results seem solid, as is their interpretation. I have only one comment below on the RG for the ladder boundary, whose treatment is rather schematic and would benefit from a more detailed explanation.

From the novelty viewpoint, I find that the work really lacks a clear motivation. The way the introduction is written and results are presented seems to indicate that this is a technical work, that expands our understanding of boundary critical behavior, but without a clear goal (such as, e.g., an experimental motivation, an open theoretical challenge, or the finding new phenomenology). Based on this, my impression is that the work better fits within the realm of SciPost Physics core.

I have few more detailed comments that I would kindly ask the authors to address:

- Sec 3.1 is presented as novel, but in fact, plenty of literature has discussed spin-1/2 ladders. The authors cite some works (35-39), but I'd strongly suggest to expand the relevant bibliography. A summary of those can also be found in Tsvelik's book "Quantum Field Theory In Condensed Matter Physics";

- the RG equations [65-73] are hard to interpret. I was wondering if some integrals of motions are present (like in the sine-Gordon case), and if that could help the interpretation presented afterwards. In general, this part of the work would strongly benefit from revisions, as its interpretation is very schematic and it's unclear whether all phenomenology is exhausted. Ideally, this could also be done by solving numerically the RG equations for sets of physically interesting coupling parameters (the full space is hard to chart).

- some typos are present (e.g., pag6, "the the VBS order"), please spell check the text;

- in the ladder discussion, I was a bit confused on whether conformal spin is conserved, or not. Can the authors please comment on this? I would find a comment useful since it may help identify which terms are most relevant here.

Requested changes

- address the comments above

- motivate clearly why the paper is suited for SciPost Physics (e.g., explaining how this meets the corresponding criteria) and revised the text correspondingly. Alternatively, the paper could be in my opinion accepted in SciPost Physics core without further major revisions.

---

## Editorial Decision

awaiting_resubmission